# Impact of COVID-19 on the Russian Labor Market: Comparative Analysis of the Physical and Informational Spread of the Coronavirus

**Dmitriy Rodionov [1], Anastasia Ivanova [1], Olga Konnikova [2] and Evgenii Konnikov [1,\*]**

[1]   Graduate School of Industrial Economics, Peter the Great St. Petersburg Polytechnic University, 195251 Saint Petersburg, Russia; rodion_dm@mail.ru (D.R.); nastyai.ivanov@yandex.ru (A.I.)

[2]   Department of Marketing, Saint-Petersburg State University of Economics, 191023 Saint Petersburg, Russia; olga.a.konnikova@gmail.com

\*   Correspondence: konnikov.evgeniy@gmail.com; Tel.: +7-9618084582

**Abstract:** The aim of the article is to investigate the impact of the new coronavirus infection on the Russian labor market and to suppose the actions to be taken to minimize negative economic consequences. The distinctiveness of this study is the differentiation of the impact of the physical and informational spread of COVID-19. The informational spread of coronavirus is measured through the dynamics of news messages related to the topic of "coronavirus" in the largest Russian media. The analysis of the average level of wages by type of economic activity, as well as the demand of employers and the number of vacancies, allow testing the hypothesis that the physical and informational spread of coronavirus caused an increase in the number of unemployed, a decrease in average wages in the studied range of economic activities, an increase in supply on the labor market, and a decrease in demand for employees. Another task of the study is to assess the dynamics of related search queries in Yandex (Russian biggest search engine), which can help to reveal the logic in the behavior of the Russian people during the pandemic as well as to understand if the Russian economy, the labor market, and society were prepared for the changes caused by the pandemic. Using a regression modeling methodology, it was found that the influence of the information environment, namely the informational spread of coronavirus, had an even greater impact on studied parameters than the physical spread. A "delay effect of physical consequences" was discovered. The conclusions obtained showed that in the conditions of wide informatization of society, it is necessary to systematically influence the physical and informational spread of coronavirus to minimize the negative consequences of the pandemic on the labor market.

**Keywords:** COVID-19; labor market; informational spread of coronavirus; information environment

## 1. Introduction

In the beginning of 2020, the world in general and Russia fell victim to a pandemic of a new virus—COVID-19. The virus changed the way of life incredibly and had a negative impact on the world economy. Instead of cooperation, it forced people into "social distancing" and "self-isolation", which led to economic stagnation and forced many employees to leave their usual places of employment, acquiring the status of "unemployed".

In the media and on social networks, panic began to grow at a rapid rate. Some industries were essentially paralyzed. Of course, it is too early to say that the crisis has gone away, however, society already has a certain amount of information about COVID-19, enabling to take certain steps to eliminate the consequences of the pandemic.

The spread of new coronavirus infections caused changes in the Russian labor market. In many respects, the forced introduction of interaction restrictions caused by the specificity of the virus led to a significant transformation of the production and consumption processes, which resulted in the disruption of many economic ties. These disruptions inevitably lead

to both a decrease in the volume of resource turnover in the economy and to its slowdown, which, in turn, resulted in a systemic slowdown in economic growth and a protracted recession. The specifics of the Russian economy lie in the fact that it belongs to the group of resource-rich and resource exporting transition economies dominated by the export of natural resources (especially oil and gas), which determines the specifics of the Russian labor market—it is largely more focused on international rather than on the domestic economy (Sadik-Zada 2020; Sadik-Zada et al. 2021). This caused the particular impact of the pandemic and the subsequent breakdown of supply chains on the Russian economy and the Russian labor market.

The urgency of the problem considered within this article lies in the need to take prompt measures to prevent the economic consequences of the pandemic. The solution to the problem is extremely relevant in the context of its global character.

Before going to the primary data set investigation, the theoretical and methodological basis of similar research interests was analyzed. Thus, it is worth highlighting a number of studies that have discovered the following key topics: technology development and innovations as one of the possible consequences of the pandemic; the impact of the information environment on society and economic processes during a pandemic; labor market changes under the influence of the coronavirus pandemic; and forecasts about possible ways of economic recovery.

To begin with, Tisdell (2020) underlined that throughout history, pandemic outbreaks have negatively affected the functioning of society, but also paradoxically cleared the way for innovations and achievements in science, economics, and political systems. As an example, he refers to the Black Death (the second-in history plague pandemic), which destroyed the division between upper and lower classes and led to the emergence of a new middle class. The author cites the idea that the coronavirus pandemic will help to accelerate the spread of Internet technologies for conferences, seminars, meetings, online training for social contacts, economic trade, and financial transactions. Forman et al. (2020) agree that the pandemic will lead to the development of technologies, as well as the increase of investments in human potential. Laing (2020) suggests that the pandemic may contribute to an increase in industrial process automation. Thus, the positive impact of the pandemic on the development of society is assumed. It was also hypothesized that due to the accelerated spread of Internet technologies, the interest of people in remote work, as well as in obtaining additional education based on Internet platforms, will only continue to increase.

Going deeper into the topic of the Internet environment, it is reasonable to consider the research by Almomani and Al-Qur'an (2020). The authors analyzed the information environment in the context of COVID-19. They argued that the introduction of restrictive measures to combat the spread of coronavirus, which resulted in social networks and online platforms becoming the most popular tool for social discussions, has had a devastating effect on the psychological state of society during the period of isolation. In the absence of any scientific evidence with reliable and comprehensive information about the new virus, rumors and fabricated news have affected the trust between the government and the public. The negative impact of the pandemic on the mental condition of people, as well as its influence on the economic sector, was reflected by Lyócsa et al. (2020). In this study, the authors conclude that stock markets around the world experienced a fall not so much because of the pandemic, but because of the fear of people about it. Researchers have proven that searching for information about the pandemic on Google correlates with price changes. Thus, during times of crisis, Google searches can be a valuable source of information for predicting stock price fluctuations. Within the framework of this topic, attention should be paid to the influence of the information environment on the real processes occurring in the economy. It should be checked how the informational spread of the coronavirus has affected supply and demand in the Russian market, and how the dynamics of average wages have changed under the influence of the information environment.

Considering studies on changes in the labor market under the influence of the pandemic, Adams-Prassl et al. (2020) emphasized the existence of a strong relationship between

the average share of tasks that employees can perform from home within their industry, and the percentage of employees who lost their jobs in the same industry during the pandemic. The greater the proportion of tasks performed from home, the less the employee is at risk of being fired. Thus, employees in the IT and related professions are least exposed to the risk of layoffs. The authors stated that the pandemic is likely to result in a large reallocation of employees. In this context, it is worth mentioning the study of Karlsson et al. (2014), who researched the impact of the 1918 Spanish flu pandemic on the economic performance of Sweden. The authors emphasized that the pandemic led to a significant increase in the number of the poor, and negatively affected the return on capital, however there was no significant impact on the change in income (neither during nor after the pandemic). The article also provided examples of other studies on the topic; the authors found that pandemics could indeed increase the number of unemployed but were unlikely to have a direct impact on the level of wages. Therefore, a hypothesis can be made that a pandemic can contribute to an increase in unemployment; nevertheless, a deeper analysis should be carried out to understand the relationship between changes in the level of wages in Russia and the coronavirus pandemic.

Further, the issue of economic recovery after the pandemic needs to be considered. Tisdell (2020) emphasized that the economic recovery from the pandemic will be slow, since in the modern world there is a high degree of interdependence between industries and countries. Douglass (2020) adds that in order to accelerate economic recovery after the pandemic, it will be necessary to make higher education programs more accessible to the general public, since higher education plays a key role in promoting socioeconomic mobility, innovation, and economic recovery. Anser et al. (2020) emphasized that prompt measures to regulate the economic and epidemiological situation are critically needed to minimize the consequences of the pandemic on a global level, since infectious diseases increase the number of the poor and slow down economic development, while an increase in health care costs contributes to economic development presented in a U-shaped relationship between economic growth and poverty reduction. In this regard, the opinion of Laing (2020) is interesting to consider. The researcher argues that, on the one hand, the economy can quickly recover, given that not a single production potential has been completely destroyed, as was in the case of economic collapses after the wars. On the other hand, periods of isolation can proceed to happen systematically over several months or years, which in the medium term will create a likely oversupply in some markets that will have to close or continue to work with low profit or loss.

All in all, within the analyzed theoretical and methodological basis, there is a consensus among scholars that the pandemic will affect many areas of society, nevertheless, it will become an impetus for the improvement and development of technologies, the emergence of innovative projects and systems. The labor market will change, there will be a redistribution of employees by industry, some industries will become less in demand. There will be an increase in the role of the Internet environment in society, which will contribute to the growth of remote work, the provision of educational services online, as well as affect the psychological state of people.

The aim of the current study is to investigate the impact of the new coronavirus infection on the Russian labor market. The distinctiveness of this study is the differentiation of the impact of the physical and informational spread of COVID-19 on the labor market in Russia, since in the age of widespread development of technologies and the importance of the information component of society cannot be ignored. Thus, one of the tasks of this study is to identify the influence of the information field on the set of elements that characterize the labor market of the Russian Federation. The informational spread of coronavirus will be measured through the dynamics of news messages related to the topic of "coronavirus". Therefore, the study of the dynamics of related search queries in Yandex (Russian biggest search engine), an analysis of the average level of wages by type of economic activity, as well as the demand of employers and the number of vacancies, will be able to give a clearer picture of the state of the economy during the outbreak of coronavirus pandemic

in Russia, in order to reflect the general wellbeing of the country's inhabitants as well as the logic of the behavior of the population in conditions of COVID-19 pandemic. This will subsequently help to shape the answer to the question of what to do to minimize the negative consequences of COVID-19 on the Russian labor market.

Regarding the rationale of the economy branches being taken into consideration within the framework of the study, it is necessary to highlight the investigation carried out by the *HeadHunter* Research Center—the largest online recruitment platform in the Russian Federation—on the most and least affected sectors of Russian economics because of the coronavirus pandemic. The research revealed that the service sector (including trading) was affected most, while such traditional sectors of the Russian economy, such as agriculture, mining, and industry, were listed as the least affected ones. The reason the service sector (including trading) was affected most is the specificity of B2C business model—because of "self-isolation" restrictions established by the government, businesses were physically incapable of selling goods and services to the customers.

The significance of the study is determined by the high share of the studied sector in the total GDP of Russia. According to Russian Federal State Statistics, the selected sector makes up more than half of the GDP of the Russian economy (see Figure 1), and in addition, more than a third of the able-bodied population of the Russian economy is employed in service sector branches taken into consideration within the framework of the study (see Figure 2).

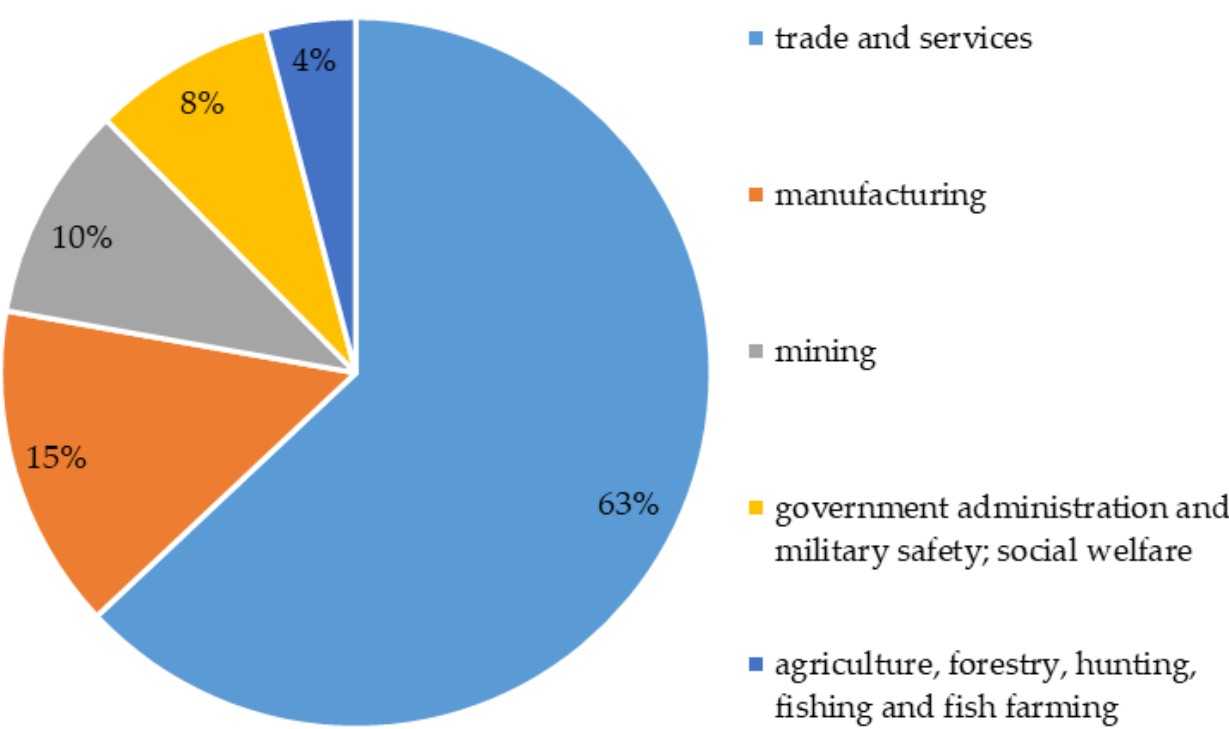

**Figure 1.** Share of sectors in the GDP of Russia.

The research methodology proposed within the paper can be universally applicable to the vast majority of sectors of the economy. However, it focusses on the specifics of the COVID-19 pandemic, and to a greater extent the impact the service sector, the reasons for which were indicated above. In this regard, the authors decided to apply the proposed methodology specifically to the service sector in order to obtain the most significant results and draw the most unambiguous applied conclusions.

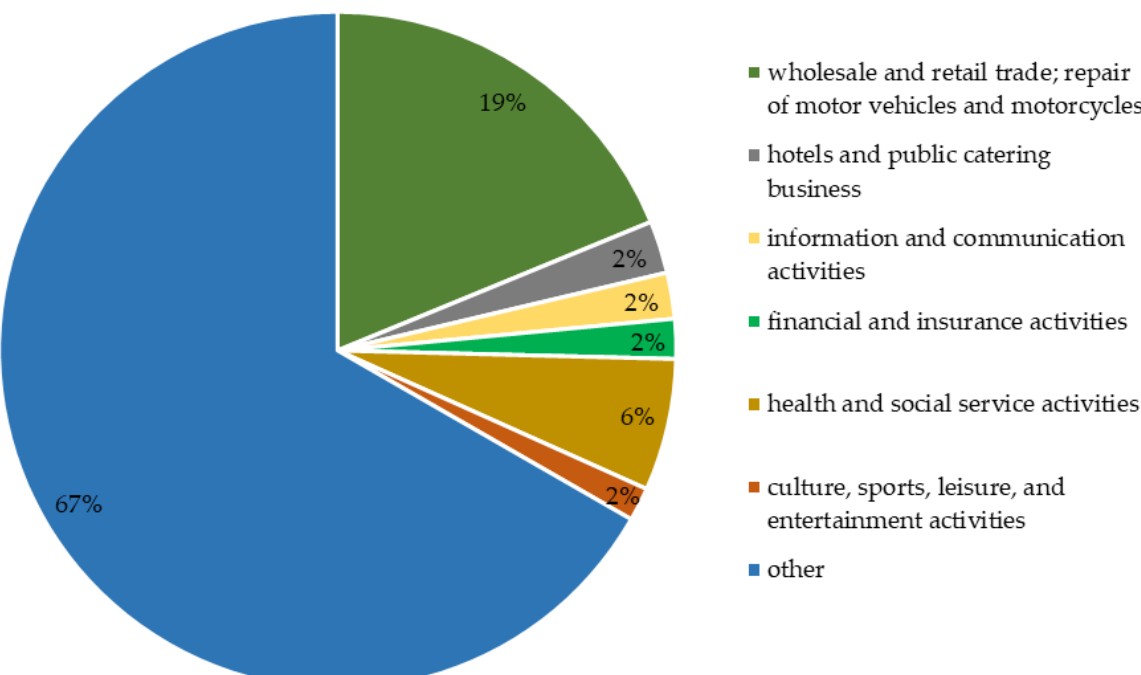

**Figure 2.** Share of the average annual numbers employed in the branches of the Russian economy (highlighting the role of the service sector).

The study itself is divided into five interrelated parts. Section 1, where the problem of the COVID-19 impact on the labor market is defined, the theoretical basis is investigated, and the investigation goals are set; Section 2, where the primary data and conceptual research model are presented and the research methodology is defined; Section 3, which includes confirmed conceptual model and systematics conclusions on the influence of factors on the set of selected indicators; Section 4, where, based on the data obtained, the hypothesized theories are taken into deeper consideration; and, finally, Section 5, where the impact of the physical and informational spread of the COVID-19 on the labor market is revealed and the recommendations for their management are given.

## 2. Materials and Methods

Before forming the conceptual model, relevant statistical data was collected. These are the data sources used for the purposes of the study:

1. *Yandex DataLens*—a service for data aggregation and analytics for identifying a number of COVID-19 cases per month in Russia;
2. The official newsgroup of *RBC (RosBusinessConsulting)*—the largest non-state media holding and a leading company working in the field of mass media and information technologies providing the latest news, the main topics of the day in politics, economics, business, and life—in *Vkontakte*, one of the most popular Russian social media, for identifying a frequency of mentioning the topic "coronavirus";
3. *Russian Federal State Statistics Service*—the governmental statistics service that collects official statistical information on social, economic, demographic, environmental and other social processes taking place in the Russian Federation for measuring the dynamics of macroeconomic indicators;
4. *Yandex Wordstat*—a service that helps to aggregate information on Yandex users' requests daily, weekly, monthly, etc., for measuring the dynamics of the search queries on the Yandex search engine.

The summary statistics are presented in Tables 1 and 2. We use some abbreviations for measures in the study: "pax" is for persons; "pcs" is for pieces; "ths. Pax" is for thousand persons; "rub". is for rubles.

**Table 1.** Summary statistics.

| Period<br>Indicator | January | February | March | April | May | June | July | August |
|---|---|---|---|---|---|---|---|---|
| The number of COVID-19 cases per month in Russia, pax. | 0 | 0 | 2337 | 104,161 | 299,345 | 242,006 | 192,132 | 155,338 |
| Frequency of mentioning the topic "coronavirus" in the *RBC (RosBusinessConsulting)* news group in Vkontakte social network, pcs | 81 | 189 | 702 | 639 | 629 | 402 | 332 | 226 |
| The number of officially registered unemployed age 15 and over, ths. pax. | 839 | 869 | 855 | 1834 | 2543 | 3152 | 3637 | 3953 |
| Total number of unemployed age 15 and over, ths. pax. | 3482 | 3425 | 3485 | 4286 | 4513 | 4606 | 4731 | 4808 |
| Average monthly nominal gross wages of employees by type of economic activity: | | | | | | | | |
| - wholesale and retail trade; repair of motor vehicles and motorcycles, rub.; | 40,685 | 40,940 | 46,359 | 42,335 | 39,136 | 42,302 | 41,258 | 39,928 |
| - hotels and public catering business, rub.; | 30,047 | 28,986 | 27,964 | 23,243 | 23,409 | 24,897 | 25,483 | 25,252 |
| - information and communication activities, rub.; | 76,215 | 78,949 | 87,942 | 92,422 | 81,754 | 82,354 | 83,716 | 80,209 |
| - financial and insurance activities, rub.; | 87,471 | 110,167 | 122,066 | 117,345 | 105,078 | 98,060 | 107,286 | 97,207 |
| - health and social service activities, rub.; | 44,565 | 43,246 | 44,957 | 42,355 | 53,147 | 53,740 | 52,501 | 49,567 |
| - culture, sports, leisure, and entertainment activities, rub. | 46,737 | 46,225 | 50,512 | 42,702 | 45,308 | 45,823 | 42,753 | 43,997 |
| Dynamics of the employers' demand for employees, declared to the bodies of the employment service, pax. | 1,464,000 | 1,496,000 | 1,493,000 | 1,346,000 | 1,385,000 | 1,518,000 | 1,639,000 | 1,692,000 |
| Workload of unemployed population per 100 announced vacancies, pax *. | 57.3 | 58.1 | 57.3 | 136.2 | 183.7 | 207.6 | 221.9 | 233.6 |
| Dynamics of the search queries on the *Yandex* search engine "удаленная работа" (remote work). | 312,209 | 286,713 | 546,777 | 774,219 | 444,288 | 351,269 | 321,044 | 304,241 |

* Workload of unemployed population per 100 announced vacancies, pax, or *tension coefficient* shows the number of unemployed citizens registered with the employment service per 100 vacancies announced by employers to the employment service. When this coefficient is more than 100, it means that the number of officially unemployed people exceeds the number of official vacancies from employers.

**Table 2.** Dynamics of the search queries on the *Yandex* search engine weekly, pcs.

| Period / Indicator | "удаленная работа" (Remote Work) | "пособие по безработице" (Unemployment Benefit) | "закрыть бизнес" (Close a Business) | "открыть бизнес" (Open a Business) | "самозанятость" (Self-Employment) | "онлайн-курсы" (Online Courses) |
|---|---|---|---|---|---|---|
| 30 December 2019–05 January 2020 | 38,867 | 73,440 | 468 | 19,622 | 14,482 | 9837 |
| 6 January 2020–12 January 2020 | 66,895 | 117,298 | 656 | 30,606 | 22,516 | 19,873 |
| 13 January 2020–19 January 2020 | 80,054 | 150,339 | 938 | 28,627 | 29,761 | 35,569 |
| 20 January 2020–26 January 2020 | 75,069 | 125,755 | 968 | 28,278 | 33,872 | 33,377 |
| 27 January 2020–02 February 2020 | 72,118 | 151,968 | 917 | 26,506 | 28,676 | 29,931 |
| 3 February 2020–9 February 2020 | 72,560 | 160,086 | 937 | 27,339 | 26,413 | 27,288 |
| 10 February 2020–16 February 2020 | 69,379 | 143,053 | 898 | 26,783 | 25,119 | 25,251 |
| 17 February 2020–23 February 2020 | 64,352 | 122,176 | 872 | 25,452 | 23,030 | 25,038 |
| 24 February 2020–01 March 2020 | 73,513 | 210,073 | 910 | 24,484 | 23,166 | 22,322 |
| 2 March 2020–8 March 2020 | 58,522 | 231,315 | 832 | 21,760 | 22,086 | 21,553 |
| 9 March 2020–15 March 2020 | 71,987 | 735,978 | 991 | 24,092 | 22,568 | 22,134 |
| 16 March 2020–22 March 2020 | 164,389 | 736,750 | 2088 | 20,977 | 21,630 | 28,199 |
| 23 March 2020–29 March 2020 | 178,123 | 513,297 | 4695 | 18,288 | 19,110 | 164,692 |
| 30 March 2020–05 April 2020 | 194,204 | 470,800 | 2597 | 15,739 | 21,262 | 404,015 |
| 6 April 2020–12 April 2020 | 160,449 | 369,894 | 1707 | 18,987 | 23,496 | 475,446 |
| 13 April 2020–19 April 2020 | 144,252 | 323,867 | 1433 | 18,855 | 26,173 | 431,281 |
| 20 April 2020–26 April 2020 | 134,205 | 375,101 | 1411 | 19,569 | 26,444 | 408,051 |
| 27 April 2020–03 May 2020 | 106,832 | 274,514 | 1177 | 18,599 | 23,225 | 289,428 |
| 4 May 2020–10 May 2020 | 93,454 | 224,172 | 1025 | 18,140 | 18,653 | 210,853 |
| 11 May 2020–17 May 2020 | 107,140 | 211,569 | 1235 | 21,283 | 33,575 | 263,145 |
| 18 May 2020–24 May 2020 | 106,719 | 226,780 | 1191 | 23,786 | 27,829 | 223,895 |
| 25 May 2020–31 May 2020 | 104,191 | 195,686 | 1276 | 55,583 | 28,257 | 306,554 |
| 1 June 2020–7 June 2020 | 93,973 | 217,116 | 1366 | 121,517 | 28,856 | 307,206 |
| 8 June 2020–14 June 2020 | 72,591 | 179,959 | 1054 | 158,718 | 21,976 | 283,556 |
| 15 June 2020–21 June 2020 | 86,380 | 162,998 | 1088 | 140,123 | 24,046 | 257,111 |
| 22 June 2020–28 June 2020 | 72,365 | 147,916 | 996 | 110,502 | 42,783 | 235,343 |
| 29 June 2020–05 July 2020 | 71,299 | 198,422 | 847 | 88,384 | 43,036 | 203,203 |
| 6 July 2020–12 July 2020 | 74,055 | 178,117 | 751 | 72,759 | 31,454 | 186,269 |
| 13 July 2020–19 July 2020 | 73,250 | 148,497 | 838 | 64,612 | 28,420 | 155,406 |
| 20 July 2020–26 July 2020 | 74,063 | 155,396 | 786 | 56,712 | 27,910 | 143,893 |
| 27 July 2020–02 August 2020 | 68,803 | 208,083 | 848 | 49,895 | 27,751 | 154,048 |
| 3 August 2020–9 August 2020 | 67,822 | 207,415 | 954 | 47,452 | 27,125 | 150,110 |
| 10 August 2020–16 August 2020 | 70,007 | 166,346 | 2167 | 42,665 | 26,362 | 143,222 |
| 17 August 2020–23 August 2020 | 70,939 | 169,481 | 1520 | 40,188 | 26,149 | 134,322 |
| 24 August 2020–30 August 2020 | 70,015 | 199,135 | 1263 | 39,370 | 28,111 | 134,357 |

Based on the analysis of the theoretical and methodological basis, a conceptual research model was formed.

The components of the conceptual model are presented in Table 3.

**Table 3.** The summary array of indicators.

| No. | Indicator | Designation | Measure | Type of Indicator | Source |
|---|---|---|---|---|---|
| 1. | The number of COVID-19 cases per month in Russia. | $X_1$ | pax | exogenous | Yandex DataLens Public (n.d.) |
| 2. | The frequency of mentioning the topic "coronavirus" in the *RBC (RosBusinessConsulting)* news group in *Vkontakte* social network. | $X_2$ | pcs | exogenous | Official News Group (n.d.) |
| 3. | The number of officially registered unemployed age 15 and over. | $Y_1$ | ths. pax | endogenous | Russian Federal State Statistics Service (n.d.) |
| 4. | Total number of unemployed age 15 and over. | $Y_2$ | ths. pax | endogenous | Russian Federal State Statistics Service (n.d.) |
| 5. | Average monthly nominal gross wages of employees by type of economic activity: | $Y_3$ | | | |
| 5.1 | *wholesale and retail trade; repair of motor vehicles and motorcycles;* | $Y_{3-1}$ | rub. | endogenous | Russian Federal State Statistics Service (n.d.) |
| 5.2 | *hotels and public catering business;* | $Y_{3-2}$ | | | |
| 5.3 | *information and communication activities;* | $Y_{3-3}$ | | | |
| 5.4 | *financial and insurance activities;* | $Y_{3-4}$ | | | |
| 5.5 | *health and social service activities;* | $Y_{3-5}$ | | | |
| 5.6 | *culture, sports, leisure, and entertainment activities.* | $Y_{3-6}$ | | | |
| 6. | Dynamics of the employers' demand for employees, declared to the bodies of the employment service. | $Y_4$ | pax | endogenous | Russian Federal State Statistics Service (n.d.) |
| 7. | The workload of the unemployed population per 100 announced vacancies. | $Y_5$ | pax | endogenous | Russian Federal State Statistics Service (n.d.) |
| 8. | Dynamics of the search queries on the *Yandex* search engine. | | | | |
| 8.1 | "удалённая работа" (remote work). | $Y_{6-1}$ | pcs | endogenous and exogenous | Yandex Wordstat (n.d.) |
| 8.2 | "пособие по безработице" (unemployment benefit). | $Y_{6-2}$ | | endogenous | |
| 8.3 | "закрыть бизнес" (close a business). | $Y_{6-3}$ | | endogenous | |
| 8.4 | "открыть бизнес" (open a business). | $Y_{6-4}$ | | endogenous | |
| 8.5 | "самозанятость" (self-employment). | $Y_{6-5}$ | | endogenous | |
| 8.6 | "онлайн-курсы" (online courses). | $Y_{6-6}$ | | endogenous | |

The relationship between the selected indicators is visualized in Figure 3.

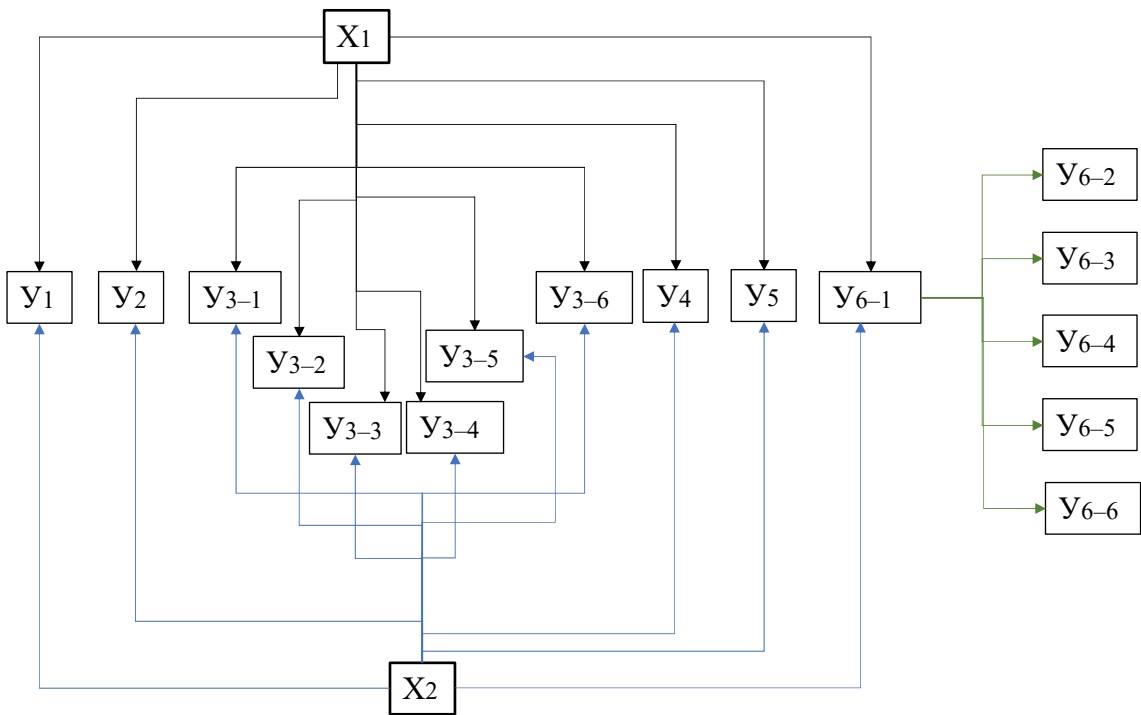

**Figure 3.** Conceptual research model.

The centroids of the conceptual model are factors $X_1$—the physical spread of coronavirus and $X_2$—the informational spread of coronavirus in Russia from January to August 2020. The initial conceptual model hypothesizes an increase in the number of unemployed under the influence of the physical and informational forms of coronavirus spread, a decrease in average wages in the studied range of economic activities, an increase in supply in the labor market, and a decrease in employers' demand for employees. It is also hypothesized that the physical and informational forms of coronavirus spread have a positive effect on the number of "remote work" searches on the Yandex search engine. Summarizing, using regression modeling, the logic of the behavior of the Russian society during the pandemic is indirectly investigated. The data obtained is mainly descriptive.

It should be noted that within the framework of the study the temporary array of statistical data is divided into three stages, corresponding to the stages of coronavirus spread in Russia from January to August 2020: the first stage is the growth of infection distribution and the absence of any governmental measures to prevent the further spread; the second stage is the peak of distribution, accompanied by severe governmental restrictions—a period of "self-isolation" for citizens, which obliged people not to leave their homes without any crucial reasons, compulsory closure of trade and service enterprises except for those selling living essentials; the third stage is the decline in the spread of COVID-19 and the mitigation of substantial part of the governmental restrictions for citizens and businesses. Statistical data for $X_1$ and $X_2$, as well as $Y_1, Y_2, ..., Y_6$ is given by month from January to August 2020. The study allows us to draw conclusions about the extent to which the Russian economy, the labor market, and society were prepared for the changes caused by the coronavirus pandemic.

As one of the basic tasks of the study is to analyze the impact of the informational spread of coronavirus on $Y_1, Y_2, ..., Y_6$, which characterize the Russian labor market from January to August 2020, the frequency of the appearance of news messages related to the topic of "coronavirus" in the *RBC Vkontakte* newsgroup was used as an indicator of the informational spread of coronavirus from January to August 2020. To conduct the study, an automated algorithm for collecting posts from the *RBC Vkontakte* group was built. Let us consider the stages of this algorithm (Rodionov et al. 2020):

1. Formation of the primary data set. Within the framework of this stage, the news array is collected in accordance with the analyzed period. The source of primary information is the largest Russian social media *Vkontakte*, namely the newsgroup of *RBC (RosBusinessConsulting)*, the largest non-state media holding and a leading company working in the field of mass media and information technologies providing the latest news, the main topics of the day in politics, economics, business, and life. Analysis of the news in this group makes it possible to understand the most relevant information messages during the spread of coronavirus in Russia. For automated parsing of information, Python 3 programming language was used. The result of this stage was eight tables containing the main news messages within the given months. To facilitate further analysis, tokens (meaningful units of text) were extracted from the news background by month. As part of the study, tokens that can give an idea of the information distribution of coronavirus in Russia were allocated.

2. Tokenization of the primary data set. To determine the dynamics of the information distribution of coronavirus in Russia, the received array of primary data was tokenized. The result of this stage is a table containing the number of tokens related to the topic "coronavirus" by month. The presence of the following tokens in the primary data set was investigated: "коронавирус" (coronavirus), "пандем" (pandem), "COVID".

The research methodology is based on the traditional regression analysis, which implies the following defining criteria:

- The presence of structural breaks/structural outliers;
- Significance of Fisher's F-test;
- P-level;
- Approximation error of the model;
- Multiple coefficient of determination $R^2$;
- Heteroscedasticity of residues;
- The presence of multicollinearity;
- Analysis of the relative coefficients of elasticity.

A potential system of regression equations would look like the following:

Potential multiple regression equations:

$$\begin{cases} Y_1 = a + b_1 X_1 + b_2 X_2 \\ Y_2 = a + b_1 X_1 + b_2 X_2 \\ Y_{3\text{-}1} = a + b_1 X_1 + b_2 X_2 \\ Y_{3\text{-}2} = a + b_1 X_1 + b_2 X_2 \\ Y_{3\text{-}2} = a + b_1 X_1 + b_2 X_2 \\ Y_{3\text{-}3} = a + b_1 X_1 + b_2 X_2 \\ Y_{3\text{-}4} = a + b_1 X_1 + b_2 X_2 \\ Y_{3\text{-}5} = a + b_1 X_1 + b_2 X_2 \\ Y_{3\text{-}6} = a + b_1 X_1 + b_2 X_2 \\ Y_4 = a + b_1 X_1 + b_2 X_2 \\ Y_5 = a + b_1 X_1 + b_2 X_2 \\ Y_{6\text{-}1} = a + b_1 X_1 + b_2 X_2 \end{cases}$$

Potential paired regression equations:

$$\begin{cases} Y_{6\text{-}2} = aY_{6\text{-}1}{}^b \\ Y_{6\text{-}3} = aY_{6\text{-}1}{}^b \\ Y_{6\text{-}5} = aY_{6\text{-}1}{}^b \\ Y_{6\text{-}6} = aY_{6\text{-}1}{}^b \end{cases}$$

To identify the impact of the physical and informational spread of coronavirus on the dynamics of indicators characterizing the labor market in Russia, we used multiple regression analysis. In this study, the confidence limit (reliability level) of the model was taken as 90%, and the permissible P-level was 10%, respectively. The chosen significance level of 10% is because the macromodel is being investigated, therefore, $Y_1, Y_2 \ldots Y_6$ are determined not only by $X_1$ and $X_2$, but also by many other factors that are not included in the described model. Thus, every time the P-level of any Y exceeds the acceptable level of 10%, the multiple regression model is being optimized.

To study changes in the numbers of search queries, such as "unemployment benefit", "close a business", "open a business", "self-employment", and "online courses" regarding the numbers of search query "remote work", we used paired regression tools to reveal the logic of people behavior during the pandemic. The array of analyzed data has been

expanded to include primary data by week, from January to August 2020. As part of the paired regression analysis, the confidence limit (reliability level) of the model was increased to 95% due to the growth of the primary data set. The methodology of the study includes the assumed square cube connection in the description of the system of equations.

### 3. Results

In accordance with the above methodology, the influence of factors on the set of elements that characterize the Russian labor market from January to August 2020 was tested. Figure 4 shows the validated conceptual model. This model allows to visualize how strongly the factors affect the studied variables.

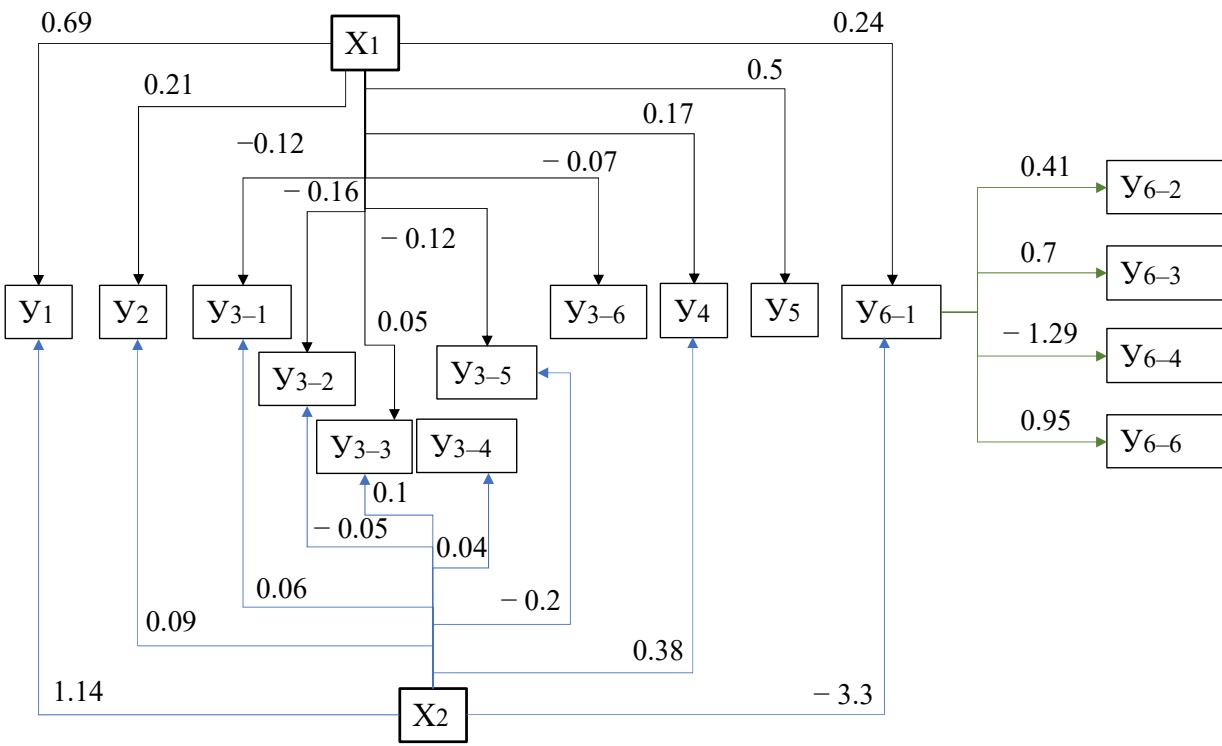

**Figure 4.** Validated conceptual model.

The validated conceptual model is described by the following system of multiple and paired regression equations (see Table 4). In Table 4 **, * indicate significance at 10% and 5%, respectively.

Based on the analysis of the relative coefficients of elasticity, the following interpretation of the main empirical findings, presented in Table 4, can be given.

*Multiple regression.* $X_1$ and $X_2$ increased the values of $Y_1$, i.e., the physical and informational spread of the coronavirus increased the number of officially registered unemployed aged 15 and over. When the physical spread of the coronavirus $X_1$ changed by 1%, the number of officially registered unemployed age 15 and over increased from 0.56 to 0.82%; when the informational spread changed by 1%, the indicator increased from 0.51 to 1.77%.

$X_1$ and $X_2$ increased the values of $Y_2$, i.e., the physical and informational spread of the coronavirus increased the total number of unemployed aged 15 and over. When the physical spread changed by 1%, the total number of unemployed aged 15 and over increased from 0.21 to 0.22%; when information spread of the coronavirus changed by 1%, the total number of unemployed aged 15 and over increased from 0.06 to 0.11%.

**Table 4.** System of multiple and paired regression equations to describe the validated conceptual model.

| Multiple Regression | | | | | | | | |
|---|---|---|---|---|---|---|---|---|
| *Equation* | *The Presence of Structural Breaks/Structural Outliers* | *Significance of Fisher's F-Test* | *P-Level (for F-Test)* | *Approximation Error of the Model* | *Multiple Coefficient of Determination $R^2$* | *The Statistical Significance for Each Coefficient* | *The Presence of Multi-collinearity* | *Analysis of the Relative Coefficients of Elasticity* |
| $Y_1 = -246.8 + 0.02 * X_1 + 19.01 * X_2 - 1.5 * 10^{-13} * X_1^3 - 0.08 * X_2^2 + 0.00008 * X_2^3$ | No structural breaks, structural outliers are extremely insignificant | 0.007 | <0.1 | 5.8% | 99.7% | $X_1$ *** $X_2$ ** $X_1^3$ *** $X_2^2$ ** $X_2^3$ ** | No multi-collinearity between $X_1$ and $X_2$ is observed | When $X_1$ changes by 1%, $Y_1$ increases from 0.56% to 0.82%. When $X_2$ changes by 1%, $Y_1$ increases from 0.51% to 1.77%. |
| $Y_2 = 3345.74 + 0.01 * X_1 + 2.8 * X_2 - 3.2 * 10^{-8} * X_1^2 - 0.016 X_2^2 + 0.00002 * X_2^3$ | Absence of structural breaks and structural outliers | 0.0002 | <0.1 | 0.2% | 99.9% | $X_1$ *** $X_2$ *** $X_1^2$ *** $X_2^2$ *** $X_2^3$ *** | No multi-collinearity between $X_1$ and $X_2$ is observed | When $X_1$ changes by 1%, $Y_2$ increases from 0.21% to 0.22%. When $X_2$ changes by 1%, $Y_2$ increases from 0.06% to 0.11%. |
| $Y_{3-1} = 40551.36 - 0.1 * X_1 + 0.000001 * X_1^2 - 2.75 * 10^{-12} * X_1^3 + 0.01 * X_2^2$ | No structural breaks, structural outliers are extremely insignificant | 0.003 | <0.1 | 1% | 98.9% | $X_1$ *** $X_1^2$ *** $X_1^3$ *** $X_2^2$ *** | No multi-collinearity between $X_1$ and $X_2$ is observed | When $X_1$ changes by 1%, $Y_{3-1}$ decreases from -0.9% to -0.15%. When $X_2$ changes by 1%, $Y_{3-1}$ increases from 0.05% to 0.07%. |
| $Y_{3-2} = 30026.7 - 0.09 * X_1 - 2.9 * X_2 + 6.43 * 10^{-7} * X_1^2 - 1.24 * 10^{-12} * X_1^3$ | Absence of structural breaks and structural outliers | 0.005 | <0.1 | 2% | 98.4% | $X_1$ ** $X_2$ * $X_1^2$ ** $X_1^3$ ** | No multi-collinearity between $X_1$ and $X_2$ is observed | When $X_1$ changes by 1%, $Y_{3-2}$ decreases from -0.09% to -0.22%. When $X_2$ changes by 1%, $Y_{3-2}$ decreases from -0.01% to -0.08%. |
| $Y_{3-3} = 74295.5 + 0.06 * X_1 + 20.9 * X_2 - 2.67 * 10^{-7} * X_1^2$ | Absence of structural breaks and structural outliers | 0.003 | <0.1 | 2% | 95.9% | $X_1$ ** $X_2$ *** $X_1^2$ *** | No multi-collinearity between $X_1$ and $X_2$ is observed | When $X_1$ changes by 1%, $Y_{3-3}$ increases from 0.02% to 0.07%. When $X_2$ changes by 1%, $Y_{3-3}$ increases from 0.08% to 0.12%. |

**Table 4.** *Cont.*

| | | Multiple Regression | | | | | | |
|---|---|---|---|---|---|---|---|---|
| *Equation* | *The Presence of Structural Breaks/Structural Outliers* | *Significance of Fisher's F-Test* | *P-Level (for F-Test)* | *Approximation Error of the Model* | *Multiple Coefficient of Determination $R^2$* | *The Statistical Significance for Each Coefficient* | *The Presence of Multi-collinearity* | *Analysis of the Relative Coefficients of Elasticity* |
| $Y_{3\text{-}4} = 96651.58 + 0.043 * X_2^2$ | Insignificant structural breaks. Structural outlier in February. | 0.03 | <0.1 | 8% | 54.7% | $X_2^2$ ** | - | When $X_2$ changes by 1%, $Y_{3\text{-}4}$ changes from −0.003% to −0.089% |
| $Y_{3-5} = 45471.8 - 0.14 * X_1 + 1.9 * 10^{-6} * X_1^2 - 4.33 * 10^{-12} * X_1^3 - 0.09 * X_2^2 + 0.00013 X_2^3$ | Absence of structural breaks and structural outliers | 0.01 | <0.1 | 1% | 99.4% | $X_1$ ** $X_1^2$ ** $X_1^3$ ** $X_2^2$ * $X_2^3$ * | No multi-collinearity between $X_1$ and $X_2$ is observed | When $X_1$ changes by 1%, $Y_{3\text{-}5}$ decreases from −0.06% to −0.19%. When $X_2$ changes by 1%, $Y_{3\text{-}5}$ decreases from −0.03% to −0.38%. |
| $Y_{3\text{-}6} = 47716.8 - 0.05 * X_1 + 1.58 * 10^{-6} * X_1^2$ | Insignificant structural breaks. Structural outlier in March. | 0.09 | <0.1 | 4% | 61.5% | $X_1$ ** $X_1^2$ * | - | When $X_1$ changes by 1%, $Y_{3\text{-}6}$ decreases from −0.02% to −0.13% |
| $Y_4 = 1214415.4 + 4263.5 * X_2 + 1.87 * 10^{-5} * X_1^2 - 5.45 * 10^{-11} * X_1^3 - 17.7 * X_2^2 + 0.017 * X_2^3$ | No structural breaks, structural outliers are extremely insignificant | 0.07 | <0.1 | 2% | 97% | $X_2$ * $X_1^2$ * $X_1^3$ * $X_2^2$ ** $X_2^3$ ** | No multi-collinearity between $X_1$ and $X_2$ is observed | When $X_1$ changes by 1%, $Y_4$ increases from 0.04% to 0.3%. When $X_2$ changes by 1%, $Y_4$ increases from 0.1% to 0.65%. |
| $Y_5 = 54.5 + 0.0012 * X_1 - 8.22 * 10^{-15} * X_1^2$ | Absence of structural breaks and structural outliers | 0.0004 | <0.1 | 14% | 95.4% | $X_1$ *** $X_1^2$ *** | - | When $X_1$ changes by 1%, $Y_5$ increases from 0.38% to 0.62% |
| $Y_{6\text{-}1} = 1347493.78 + 1.71 * X_1 - 10367.4 * X_2 - 8.57 * 10^{-6} * X_1^2 + 0.71 * X_1^3 + 29.6 * X_2^2 - 0.7 * X_2^3$ | Absence of structural breaks and structural outliers | 0.002 | <0.1 | 0.15% | 99.9% | $X_1$ *** $X_2$ *** $X_1^2$ *** $X_1^3$ *** $X_2^2$ *** $X_2^3$ *** | No multi-collinearity between $X_1$ and $X_2$ is observed | When $X_1$ changes by 1%, $Y_{6\text{-}1}$ increases from 0.23% to 0.25%. When $X_2$ changes by 1%, $Y_{6\text{-}1}$ decreases from −3.21% to −3.39%. |

**Table 4.** *Cont.*

| | | | | | | | | | |
|---|---|---|---|---|---|---|---|---|---|
| | | | | **Multiple Regression** | | | | | |
| *Equation* | *The Presence of Structural Breaks/Structural Outliers* | *Significance of Fisher's F-Test* | *P-Level (for F-Test)* | *Approximation Error of the Model* | *Multiple Coefficient of Determination $R^2$* | *The Statistical Significance for Each Coefficient* | *The Presence of Multi-collinearity* | *Analysis of the Relative Coefficients of Elasticity* |
| | | | | **Paired regression** | | | | | |
| $Y_{6-2} = 1.045 * Y_{6-1}^{1.075}$ | Insignificant structural breaks. Structural outlier in March. | <0.05 | <0.05 | 2.7% | 57% | $Y_{6-1}$ *** | - | When $Y_{6-1}$ changes by 1%, $Y_{6-2}$ increases from 0.35% to 0.47% |
| $Y_{6-3} = 0.0215 * Y_{6-1}^{0.9572}$ | Insignificant structural breaks. Structural outlier in the end of March. | <0.05 | <0.05 | 3.6% | 63.6% | $Y_{6-1}$ *** | - | When $Y_{6-1}$ changes by 1%, $Y_{6-3}$ increases from 61% to 79% |
| $Y_{6-4} = 5 * 10^7 * Y_{6-1}^{-0.631}$ | Insignificant structural breaks. Structural outlier in June. | <0.05 | <0.05 | 5.9% | 11.6% | $Y_{6-1}$ ** | - | When $Y_{6-1}$ changes by 1%, $Y_{6-4}$ decreases from −0.68% to −1.9% |
| | | | | **No connection between $Y_{6-5}$ и $Y_{6-1}$ (P-level > 0.05)** | | | | | |
| $Y_{6-6} = 0.0001 * Y_{6-1}^{1.8131}$ | Insignificant structural breaks. Structural outliers in March and June. | <0.05 | <0.05 | 8.24% | 30.9% | $Y_{6-1}$ *** | - | When $Y_{6-1}$ changes by 1%, $Y_{6-6}$ increases from 0.7% to 1.2% |

***, **, and * indicate statistical significance at the 1%, 5%, and 10% levels, respectively. Potential heteroscedasticity is observed in all equations. Heteroskedasticity and autocorrelation consistent standard errors were used to provide an estimate of the covariance matrix of the parameters of a regression-type models (Paz 2022).

$X_1$ and $X_2$ had a multidirectional effect on $Y_{3-1}$, i.e., the physical spread of the coronavirus decreased the average monthly nominal gross wages of employees in wholesale and retail trade, repair of motor vehicles, and motorcycles; while the informational spread of the coronavirus increased the indicator. When the physical spread changed by 1%, the average monthly nominal gross wages decreased from $-0.9$ to $-0.15$%; when the information spread of the coronavirus changed by 1%, the indicator increased from 0.05% to 0.07%.

$X_1$ and $X_2$ decreased the values of $Y_{3-2}$, i.e., the physical and informational spread of the coronavirus decreased the average monthly nominal gross wages of employees in hotels and public catering businesses. When the physical spread changed by 1%, the average monthly nominal gross wages of employees in hotels and public catering businesses decreased from $-0.09$ to $-0.22$%; when the information spread of the coronavirus changed by 1%, the indicator decreased from $-0.01$ to $-0.08$%.

$X_1$ and $X_2$ increased the values of $Y_{3-3}$, i.e., the physical and informational spread of the coronavirus increased the average monthly nominal gross wages of employees in information and communication activities. When the physical spread of coronavirus $X_1$ changed by 1%, the average monthly nominal gross wages of employees in information and communication activities increased from 0.02 to 0.07%; when the information spread of the coronavirus changed by 1%, the indicator increased from 0.08 to 0.12%.

$X_2$ increased the values of $Y_{3-4}$, i.e., when the information spread of the coronavirus changed by 1%, the average monthly nominal gross wages of employees in financial and insurance activities increased from $-0.003$ to 0.089%.

$X_1$ and $X_2$ decreased the values of $Y_{3-5}$, i.e., the physical and informational spread of the coronavirus decreased the average monthly nominal gross wages of employees in health and social service activities. When the physical spread of coronavirus $X_1$ changed by 1%, the average monthly nominal gross wages of employees in health and social service activities decreased from $-0.06$ to $-0.19$%; when the information spread, $X_2$ changed by 1%, the indicator decreased from $-0.03$ to $-0.38$%.

$X_1$ decreased the values of $Y_{3-6}$, i.e., when the physical spread of the coronavirus changed by 1%, the average monthly nominal gross wages of employees in culture, sports, leisure, and entertainment activities decreased from $-0.02$ to $-0.13$%.

$X_1$ and $X_2$ increased the values of $Y_4$, i.e., the physical and informational spread of the coronavirus increased the employers' demand for employees, declared to the bodies of the employment service. When the physical spread of coronavirus $X_1$ changed by 1%, the employers' need for employees, declared to the employment service bodies, increased from 0.04 to 0.3%; when the informational spread changed by 1%, the indicator increased from 0.1 to 0.65%.

$X_1$ increased the values of $Y_5$, i.e., the physical spread of the coronavirus increased the workload of the unemployed population per 100 announced vacancies. When the physical spread of the coronavirus changed by 1%, the indicator increased from 0.38 to 0.62%.

$X_1$ and $X_2$ had a multidirectional effect on $Y_{6-1}$, i.e., the physical spread of the coronavirus increased the number of search queries on the Yandex search engine "удаленная работа" (remote work), while the informational spread of coronavirus decreased the indicator. When the physical spread of the coronavirus changed by 1%, the number of search queries increased from 0.23 to 0.25%; when the information spread of the coronavirus changed by 1%, the indicator decreased from $-3.21$ to $-3.39$%.

*Paired regression.* $Y_{6-1}$ increased the values of $Y_{6-2}$, i.e., search queries on the Yandex search engine "удаленная работа" (remote work) increased the number of search queries "пособие по безработице" (unemployment benefit). When search queries on the Yandex search engine "удаленная работа" (remote work) changed by 1%, search queries "пособие по безработице" (unemployment benefit) increased from 0.35 to 0.47%.

$Y_{6-1}$ increased the values of $Y_{6-3}$, i.e., search queries on the Yandex search engine "удаленная работа" (remote work) increased the number of search queries "закрыть бизнес" (close a business). When search queries on the Yandex search engine "удаленная

работа" (remote work) changed by 1%, the search queries "закрыть бизнес" (close a business) increased from 61 to 79%.

$Y_{6-1}$ decreased the values of $Y_{6-4}$, i.e., search queries on the Yandex search engine "удаленная работа" (remote work) decreased the number of search queries "открыть бизнес" (open a business). When search queries on the Yandex search engine "удаленная работа" (remote work) changed by 1%, the search queries "открыть бизнес" (open a business) decreased from −0.68 to −1.9%.

$Y_{6-1}$ increased the values of $Y_{6-6}$, i.e., search queries on the Yandex search engine "удаленная работа" (remote work) increased the number of search queries "онлайн-курсы" (online courses). When search queries on the Yandex search engine "удаленная работа" (remote work) changed by 1%, the search queries "онлайн-курсы" (online courses) increased from 0.7 to 1.2%.

Thus, the following systemic conclusions can be drawn:

1.  The spread of the new coronavirus infection has indeed contributed to an increase in the unemployment rate in the Russian labor market (see equations $Y_1$ and $Y_2$).
2.  The impact of the physical and informational spread of coronavirus on the dynamics of the average monthly nominal gross wages of employees by type of economic activity can be described as follows:
    -   for the section "Wholesale and retail trade; repair of motor vehicles and motorcycles" wages increased in the period January to February 2020, which may be primarily due to the fear that arose in the Internet environment in anticipation of the spread of a new infection, then fell from March to May 2020, after which it stabilized. Interestingly, the physical spread of coronavirus negatively affected the average wage level, while the informational spread did it in a positive way (see equation $Y_{3-1}$).
    -   for the section "Hotels and public catering business", the average level of wages demonstrated a decrease from January to April 2020. Starting from May, the analyzed values gradually levelled out (see equation $Y_{3-2}$).
    -   for the section "Information and communication activities", an increase in the average level of wages happened in the period from January to April 2020, since the physical and informational spread of coronavirus contributed to the intensification of activities related to the creation of content of various forms; in April to May the impact of coronavirus on the analyzer started to decline (see equation $Y_{3-3}$).
    -   for the section "Financial and insurance activities", the model demonstrated an increase in average wages from January to March 2020. Further, from March to August, there was a negative dynamic in average wages. It is worth mentioning that in the process of optimization by the P-level criterion, the $X_1$ indicator (physical spread of coronavirus) was completely excluded from this model, see equation $Y_{3-4}$.
    -   the section "Health and social service activities" was characterized by an unstable change in the average level of wages from January to April 2020, however, since April, the average level of wages increased rapidly and reached its peak values in June 2020 (see equation $Y_{3-5}$).
    -   the section "Culture, sports, leisure, and entertainment activities" was characterized by negative impact of coronavirus spread on average wages, and in this case, we are talking only about the physical spread of coronavirus, since according to the P-level criterion, indicator $X_2$ (informational spread of coronavirus) was completely excluded from the model, see equation $Y_{3-6}$.
3.  There was a gradual decrease in the demand of employers for employees from January to April 2020. Further, the physical and informational spread of coronavirus, after reaching its peak values in April–May, begins to invariably increase the demand of employers for employees; and it is noteworthy that the informational spread of coronavirus had the greater impact on $Y_4$ (see equation $Y_4$).

4.  With regard to the impact of coronavirus spread on the workload of unemployed population per 100 announced vacancies from January to August 2020, the analyzed indicator increased sharply since March 2020. Such dynamics speak about the increase in the number of citizens wishing to start working, and an increase in competition for every vacancy. In this model, $X_2$ (informational spread of coronavirus) was completely excluded according to the P-level criterion, that is, the described dependence was a consequence of the physical spread of coronavirus only (see equation $Y_5$).

5.  The influence of the spread of coronavirus on the dynamics of search query "remote work" on the Yandex search engine from January to August 2020 was also proved. The increase in the number of search queries from February to April may be due to the physical spread of coronavirus in Russia and the lack of sufficient information about the new disease—people were afraid of its potential impact on the labor market, therefore, they were looking for ways to make money in the new reality. In general, it was found that the physical spread of coronavirus increased the number of related search queries on the Yandex search engine from January to August 2020, while informational spread of coronavirus, on the contrary, reduced the number of these search queries, and it was the latter that had the greatest impact on $Y_{6-1}$, see equation $Y_{6-1}$.

6.  As for the dynamics of related search queries on the Yandex search engine, the following dependencies were identified:

    -   analysis of the dynamics of the search query "unemployment benefit" demonstrated the presence of a structural outlier in March 2020, which is quite natural: it was in March that the first cases of COVID-19 infection were registered in Russia, and there was a massive closure of enterprises due to the introduction of severe restrictive measures, people literally remained without means of livelihood, therefore, they showed a particular interest in possible support from the state in the form of unemployment benefits. Overall, the change in the number of searches for "remote work" contributed to an increase in the number of searches for "unemployment benefit" in March 2020; starting from April, the number of searches for "unemployment benefit" gradually began to decline (see equation $Y_{6-2}$).

    -   analysis of the dynamics of the search query "close a business" on the Yandex search engine also demonstrated the presence of a structural outlier in March 2020: as mentioned earlier, in early March, the first cases of COVID-19 infection were registered in Russia, there was a consistent introduction of severe restrictive measures, and already at the end of the month, many enterprises were closed due to unprofitability and the impossibility of further functioning, thus, people began to especially actively search for relevant information on the Internet. The structural outlier was also observed in August 2020; this point can primarily be explained by the seasonal nature of some types of business (see equation $Y_{6-3}$).

    -   analysis of the dynamics of the search query "open a business" on the Yandex search engine shows the presence of a structural outlier in June 2020, which can also be explained by seasonal specifics, the beginning of the tourist season, as well as the gradual removal of some restrictions in Russia. However, the model shows a negative trend in the "open a business" search query over the analyzed period. $Y_{6-1}$ has a negative impact on the dynamics of $Y_{6-4}$, meaning that the dynamics of the search query "remote work" invariably reduce the number of search queries "open a business" (see equation $Y_{6-4}$).

    -   analysis of the dynamics of the search query "self-employment" showed no connection with the dynamics of the search query "remote work"; this is the only model that has not been confirmed.

    -   analysis of the dynamics of the search query "online courses" shows the presence of structural outliers in March and June 2020. The structural outlier in March may be associated with the tense situation in connection with the spread of COVID-19,

the anxiety of citizens was accompanied by a decrease in interest in online courses and an increase in interest in other types of information (in particular, the search for information on measures of support from the state during the pandemic). Structural outliers in June may be related to the onset of the holiday season (see equation $Y_{6-6}$).

## 4. Discussion

Based on the data obtained, it can be concluded that the presence of most of the hypothesized links was confirmed.

Thus, the impact of the physical and informational spread of coronavirus from January to August 2020 on the following elements of the Russian labor market was confirmed: number of officially registered unemployed; total number of unemployed; average monthly nominal gross wages by such types of economic activity as "Wholesale and retail trade, repair of motor vehicles and motorcycles"; "Hotels and public catering business"; "Information and communication activities"; "Health and social service activities"; dynamics of the employers' demand for employees, declared to the bodies of the employment service; dynamics of the search query "remote work" on the Yandex search engine. As part of the paired regression analysis, the influence of the numbers of the search query "remote work" on the number of such search queries as "unemployment benefit", "close a business", "open a business", and "online courses" was also confirmed.

However, several hypotheses have not been statistically confirmed. The study found that the physical spread of coronavirus did not affect the average monthly nominal wages for the type of economic activity "Financial and insurance activities"—this indicator is only influenced by the informational spread of coronavirus. The influence of the informational spread of coronavirus on the average monthly nominal gross wages for the type of economic activity "Culture, sports, leisure, and entertainment activities", as well as on the workload of the unemployed population per 100 declared vacancies, also has not been confirmed—these indicators are only influenced by the physical spread of coronavirus. As part of the paired regression analysis, we had to refute the hypothesis that "remote work" searches influence "self-employment" searches. The assumption that the pandemic contributed to a decrease in employers' demand for employees also has not been confirmed.

The analysis of the statistical dataset made it possible to clarify and supplement the theoretical basis of this study. Thus, the impact of the coronavirus pandemic in Russia on the growth of unemployment was confirmed. In the paper, the focus is set on the employees and the average rates of their wages. Since citizens and income are broader indicators, commenting on the statement that there is no direct effect of the pandemic on the average income of citizens would be incorrect. Nevertheless, the analysis established a clear relationship between the spread of coronavirus infection and changes in the average wages of employees. It should be especially noted the specificity of the influence of the informational and physical forms of coronavirus spread on the analyzed indicator. In most cases, the change in average wages was influenced by both forms of the spread of coronavirus infection, and only in two cases out of seven was it identified that the influence of only one form of coronavirus spread on the analyzed indicator.

In some cases, the multidirectional influence of the informational and physical forms of coronavirus spread on the average level of wages has been revealed. It can be concluded that there are blocks of professional activities that are more sensitive to impulses coming to the real world from the information world.

The assumptions that employees in the IT professions are less susceptible to the negative effects of epidemiological outbreaks have been partly confirmed. Nevertheless, it is impossible to say unequivocally that the pandemic led to a large redistribution of employees in the labor market in Russia: as part of the analysis of paired regressions, it was found that the majority of Russians quickly adapted to the changing conditions during a difficult epidemiological situation and continue to work in their specialization. The most

significant change was not a change in the field of professional activity, but a change in the form of employment: from freelance to hired labor.

Within the framework of the study, the block devoted to the study of the dynamics of changes in related search queries regarding the search query "remote work" should be highlighted. The study of these dependencies indirectly allowed to reveal the logic of the behavior of the Russian society during the spread of the new coronavirus infection. Of course, it cannot be said that the study of paired regressions fully describes the logic of people's behavior during the pandemic (to establish such connections it is necessary to analyze a much larger array of search queries), nevertheless, it is possible to draw conclusions how much society was ready for the changes that took place, to what extent citizens were ready to change their behavior patterns in a crisis.

The dynamics of the numbers of the search query "remote work" had a 70% impact on the dynamics of the numbers of the search query "close a business", which is the most significant indicator of elasticity coefficient in the framework of all validated regression equations. This means that most of the people who showed interest in working from home were most likely forced to stop their entrepreneurial activity, which means that they had to move from the model of working "for oneself" to the model of working "for someone else"; someone whose business has not been affected so greatly by the spread of the coronavirus infection and consistently imposed restrictions. It is noteworthy that the more people are interested in remote work, the less they show interest in starting their own business.

In addition, within the framework of paired regression, we analyzed the dynamics of changes in the search query "online courses" in relation to the query "remote work". The coefficient of elasticity for this variable showed values of less than 1%. Thus, it can be assumed that the bulk of society interested in switching to remote work during the spread of the new coronavirus infection in Russia was generally satisfied with the area of professional activity that they had been engaged in before the onset of the pandemic. A smaller proportion of citizens nevertheless decided to master new skills necessary to continue working in the context of the "coronavirus" reality.

The elasticity coefficient for the search query " unemployment benefit" as part of testing the paired regression model showed the lowest values −0.41%. Thus, people were less interested in support from the state in the context of remote employment. This logic of behavior can be explained by the fact that during the period of the spread of COVID-19 and the introduction of restrictive measures, there was an opinion in the Russian society that remote work was nothing more than a new form of employment and it is quite equivalent to the usual way of working, which means that can also provide an acceptable level of earnings.

Essentially, the conducted research confirms the idea that along with the objective world, there is also the information world, the significance of which cannot be ignored since it exerts its influence on the totality of processes and phenomena in the objective world. Moreover, as part of the study, it was found that the influence of the information environment, namely the informational spread of coronavirus, had an even greater impact than the physical spread.

As part of the analysis of multiple regressions, the "delay effect of physical consequences" was discovered. It means that the consequences of the informational spread of coronavirus, in some cases, brought a return to the transformation of the objective world earlier than the consequences of the physical spread of coronavirus.

Within the framework of the study, it should also be noted that the information environment does not always have an impact with the same directionality as the objective environment. In several cases, the inverse dependences of the informational and physical spread of coronavirus on the elements that characterize the labor market were revealed. This, among other things, can be explained by the mentioned "delay effect of physical consequences" phenomenon.

Thus, the physical spread of coronavirus has a direct impact on the elements of the labor market, and informational spread, in addition to direct impact, also has an indirect impact on the transformation of the external environment, as well as the behavior of society.

Thus, it seems logical to conclude that the information world has a transformational impact on the external environment, therefore, the physical and informational forms of the spread of coronavirus are related to each other, which is why a systematic management of these factors in order to minimize the negative impact of COVID-19 on the labor market in Russia is needed.

In the context of widespread informatization of society, ignoring fluctuations in the information environment is unreasonable because it is not only capable of influencing the processes of the objective world, but also can exert a much stronger influence than material factors. In this regard, the speed and the quality of information flow related to COVID-19 should be paid attention first. It is the quality of information related to the topic of coronavirus that can become a defining issue in minimizing negative impacts on the labor market, as well as the psychoemotional state of people. The more the Internet environment is saturated with "empty" information that does not give the user an objective assessment of the current situation, the faster the growth of uncontrolled, impulsive actions of the people will be. That firstly happened in the first quarter of 2020. The fear that struck the country during the onset of the pandemic contributed to the growth of social tension and confusion, which, in turn, impacted on the real processes and phenomena of the physical world (the economic state of the labor market in Russia).

## 5. Conclusions

The findings obtained lead to the conclusion that the coronavirus pandemic contributed to an increase in the number of unemployed and had a decisive impact on the change in average wages. It was also revealed that in the conditions of wide informatization of society, special attention should be paid to the information environment. The so-called "delay effect of physical consequences" was identified: the consequences of the informational spread of coronavirus, in some cases, brought a return to the transformation of the phenomena of the material world earlier than the consequences of the physical spread of coronavirus, which had an impact on the processes of the material world with a certain time delay. The study also established the rapid adaptation of Russian society to the new conditions of remote employment.

It is necessary to systematically influence the physical and informational spread of coronavirus in order to minimize the negative consequences of the pandemic on the labor market. Methods for the implementation of this impact can be conditional, in accordance with the basic division of distribution types into two categories—physical and informational. The physical methods should primarily include the strengthening of the predictive properties of the development of medical systems. This strengthening can be manifested in an increase in the frequency of updating the database of medicines and a partial transition of medical organizations to notification financing, which implies the provision of funds requested by a medical organization from the federal or regional budget without examination of the application and outside the established schedule. Information methods primarily imply the formation of a directed information (primarily news) stream, characterized by the stabilization of the coronavirus topic within the general information context. In the context of Russian reality, minimizing "empty" news blocks, providing only meaningful information related to the coronavirus, and excluding the possibility of manipulating the public for the sake of personal political and economic interests, can contribute to the early stabilization of the Russian economy and the labor market from the consequences of COVID-19.

The study can be improved by investigating COVID-19's impact on a wider range of areas of professional activities and taking into consideration more complex mechanisms, related to government policies, lockdowns and etc., influencing unemployment and other macroeconomic indicators. Moreover, a more profound exploration of citizens' psycho-

emotional state of mind and a broader selection of search queries on the Yandex search engine during the pandemic period would increase the significance of the following studies.

In spite of the fact that this study examines the impact of COVID-19 on the most affected sector of the Russian economy, the significance of the study to the whole Russian labor market is beyond question due to the high share of the selected sector in the total GDP of Russia and the high share of the able-bodied population of the Russian economy employed in the selected service sector branches.

The study confirms that coronavirus' physical spread directly impacted on elements of the Russian labor market, and informational spread indirectly impacted on the transformation of the external environment, as well as the behavior of society, which leads to the necessity of modeling information flows to minimize negative economic effects on the Russian labor market. That means that a similar influence may take place on international labor markets, which determines the possibility to apply the results of this study by authors outside Russia.

**Author Contributions:** Conceptualization, D.R.; methodology, A.I., E.K. and O.K.; software, A.I.; validation, A.I.; formal analysis, A.I.; investigation, A.I., E.K. and O.K.; resources, D.R.; data curation, O.K.; writing—original draft preparation, A.I., E.K. and O.K.; writing—review and editing, D.R. and A.I.; visualization, A.I.; supervision, D.R.; project administration, E.K.; funding acquisition, D.R. All authors have read and agreed to the published version of the manuscript.

**Funding:** This research was funded by Ministry of Science and Higher Education of the Russian Federation under the strategic academic leadership program 'Priority 2030' [(Agreement 075-15-2021-1333 dated 30 September 2021)].

**Institutional Review Board Statement:** Not applicable.

**Informed Consent Statement:** Not applicable.

**Data Availability Statement:** All used data are placed in the public domain and are available at the links: https://rosstat.gov.ru/ (accessed on 25 May 2022); https://datalens.yandex/7o7is1q6ikh23?tab=X1&utm_source=cbmain (accessed on 25 May 2022); https://wordstat.yandex.ru (accessed on 25 May 2022).

**Acknowledgments:** The research is partially funded by the Ministry of Science and Higher Education of the Russian Federation under the strategic academic leadership program 'Priority 2030' (Agreement 075-15-2021-1333 dated 30 September 2021).

**Conflicts of Interest:** The authors declare no conflict of interest.

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
