# Peer review of "Impact of COVID-19 on the Russian Labor Market: Comparative Analysis of the Physical and Informational Spread of the Coronavirus"

_economies, doi:10.3390/economies10060136_

Round 1

Reviewer 1 Report

A brief summary 

This manuscript aims to examine the impact of COVID 19 on the Russian labor market, so that authors can recommend actions that can minimize negative consequences on the economy. The hypotheses tested are whether the physical and informational spread of coronavirus causes a low demand and high supply of labor, decrease in wages, and increase in unemployed.

The study also attempts to understand the logic in the behavior of Russian people during the pandemic, and if the Russian economy, and Russian people were prepared for the changes caused by the pandemic. The study found that more than the physical impact, was not informational influence of the coronavirus. The study recommends that it is important systematically influence both physical and informational spread of the coronavirus.

Broad comments 

Strengths:

This manuscripts’ strengths are that it is timely, pertinent, and relevant. As we all know, the COVID 19 pandemic has caused more psychological and emotional damage than even the physical consequences. This influence of COVID on our thinking and behavior warrants a study such as the one the authors have written. Psyco- emotional changes take time to wean away and usually stay with the society for a longer period of time. Therefore, the negative consequences on the economy may have a long-lasting effect on the labor market. The urgency of the above problem requires us to come up with solutions that can be used to alleviate the negative consequences on the economy.

The study highlights the potential positive effect of the pandemic, in that, they would be a spread of use of Internet technologies for meeting platforms. And the interest in social technologies, as well as remote work will continue to enhance.

Weaknesses:

During the pandemic, there was an extreme demand of healthcare workers, employees that work in morgue, grocery store workers, IT professionals that work on the remote technology, pharmaceutical workers and scientists who develop vaccines and do research, laboratory workers, ultrasound technicians, radiology technicians, police, etc.

The study’s framework does not account for those professions that were high demand and low in supply. I think  this is a major flaw in the study.

The use of multiple and paired regression equations is robust. However, I suggest the authors use pearson correlation coefficients and T tests before running the regression equations.

Specific comments

Line number 9 – use the word ‘distinctiveness’ instead.

Lines 68-70- “Tisdell (2020) explains that throughout 68 history, pandemic outbreaks have not only negatively affected the functioning of society, 69 but also cleared the way for innovations and achievements in science, economics, and po- 70 litical systems”.  There are two tones in the above sentence. In the same sentence, the author talks about a negative effect of pandemics, as well as positive effects of a pandemic. Therefore, this is very confusing.

Lines 470-472-“The horror 470 that declared the country contributed to the growth of social tension and confusion, 471 which, in turn, had an impact on the real processes and phenomena of the physical world 472 (the economic state of the labor market in Russia)”. This sentence is confusing. Do you mean that the horror that was created during the onset of the pandemic?

Lines 473-476- “Improving the quality of information 473 and reducing the number of overlapping information messages (which do not introduce 474 fundamentally new, significant information into the general information background re- 475 lated to the coronavirus topic) can help to cleanse the information environment from data 476 that increase the level of social anxiety”. I suggest not making claims that you have not measured. The study does not measure social anxiety and therefore cleaning the information environment, to  reduce the social anxiety is  a far-fetched argument.

Author Response

We are truly grateful for your approving feedback on the article and the advice given to strengthen it!

We should admit that within the framework of the study we really don’t pay primary attention to professions you write about. Anyway, it should be empathized that some of these professions are indirectly investigated in the study according to the list of selected branches (for example, healthcare workers, IT professionals). The reason is the focus of the study is on that branches of the Russian economy, which experienced the most severe impact caused by the coronavirus pandemic.

By the way, thank you for your attentiveness, the usage of word “distinctive” instead of “distinctiveness” is really our flaw, we have corrected it according to your advice. Moving on, the sentence, starting with “Tisdell (2020) explains that throughout history…” has also been transformed in order not to confuse the reader. In addition, in the sentence starting with “The horror that declared the country contributed to the growth of social tension and confusion…” we really meant the fear that stroke the country on the onset of the pandemic, we have made necessary modifications to make the narration clearer. Finally, we agree that making claims about what was not measured is really a shortcoming, since information was not statistically measured and is more of logical deduction, we have excluded it from the article.

We take this opportunity of thanking you one more time for your attentive assistance and contributing so much to enhancing the study!

Reviewer 2 Report

The authors must present clearly their regression results. I see only discussion on the regression analysis but see no output. Shorten the abstract. Increase the readability of the paper.  

Author Response

We are truly grateful for your feedback on the article and the advice given to strengthen it!

According to your recommendation, we have improved the presentation of the regression results. Current version of the article includes leaner presentation of the regression results with meaningful interpretation just under Table 4 (System of multiple and paired regression equations to describe the validated conceptual model).

Moreover, we have shortened the abstract, now it looks leaner. By the way, the readability of the paper has been increased by making the narration more logically structured and accompanied by statistic data and visualizations.

We take this opportunity of thanking you one more time for your attentive assistance and contributing so much to enhancing the study!

Reviewer 3 Report

This paper examines the impact of COVID-10 on labor market outcomes in Russia. While the topic itself is relevant and up-to-date, the contributions of the study remain unclear, for the following reasons.

  • Although the paper employs a quantitative approach by using regression models, there is no explanation on data. I have no idea on how the authors collected the data, how many samples they have, what the averages are, etc. The authors are expected to demonstrate, in one independent section, about such explanation about the data together with at least a table of summary statistics.

  • It is difficult to interpret the main empirical findings shown in Table 2, because there is no explanation on the analytical framework. For example, regarding Y1, the authors conclude “When X1 changes by 1%, Y1 increases from 0.56% to 0.82%,” but it is not clear how we should interpret in this way. Did authors use natural logarithm in their analysis? Where did they get such numbers as 0.56 or 0.82? Without clear explanation on how to interpret their empirical findings, the readers cannot identify the paper’s contributions. The same comments apply to all the variables shown in Table 2.

Author Response

We are truly grateful for your profound review on the paper!

Indeed, the study employs a quantitative approach by using regression models. We should outline that before forming regression models, some relevant data was collected, which is now presented in Tables 1-2.

Moreover, the list of the sources we used is now described in the very beginning of "Materials and methods" section. In addition, since there are some abbreviations for measures in Tables, there are their interpretations there, just above Table 1, which is aimed at making the study more readable.

Moving on, we admit our flaw, concerning clear interpretation of the main empirical findings. In order to enhance the study and give the readers opportunity to identify the paper’s contributions, we have improved the presentation of the regression results. Now there is their meaningful interpretation there under Table 4 (System of multiple and paired regression equations to describe the validated conceptual model).

We take this opportunity of thanking you one more time for your attentive assistance and contributing so much to enhancing the study!

Reviewer 4 Report

In the review I would like to focus on three problems.

I. The author/authors can choose the sectors and branches that they study. However, the Reader needs to know the rationale behind their choice. Readers need to know about the scale and scope of the sectors and branches studied in the Russian economy and, consequently, to what scale and scope the study results and conclusions can be generalized to one or several sectors of the Russian economy and to how many branches of the Russian economy. For this purpose it is necessary to present in a table or tables as follows:

1)share of sectors in the studied/researched sector(s) of the economy in the GDP of Russia, assuming, all sectors in Russia equal 100%,

2)share of employment in the studied/researched sector(s) of the economy, assuming, all sectors in Russia equal 100%,

3)unemployment rate in the sectors in the surveyed/researched sector(s) of the economy, assuming all sectors in Russia equal 100%,

4)average wage in the sectors in the studied/researched sector/sectors of the Russian economy, assuming, average wage in the Russian economy = 100,

5)share of the studied branches in the GDP of Russia, assuming, all branches equal 100%,

6)share of employment in the studied branches of the economy, assuming, all branches in Russia equal 100%,

7)unemployment rate in the studied branches of the economy, assuming all branches in Russia equal 100%,

8)average wage in the studied branches of the Russian economy, assuming, average wage in the Russian economy = 100.

Of course, I leave the presentation of the division of the Russian economy into sectors to the Author/Author. One can choose one of the two suggested divisions:

1.according to the breakdown of the economy into five sectors: industry, services, agriculture, trade and construction or

2.according to the breakdown of the economy into three sectors: industry (including construction, services (including non-market services) or

3.any other available breakdown of the Russian economy by sector available from international or Russian statistics.

The branch classification used is the one commonly used in research and can be left.

The author/authors in the study did not include all sectors of the Russian economy, especially the broadly defined industrial sector (including: heavy industry, chemical industry, processing, food industry, construction and mining, etc.) and agriculture. The study basically focused on the service sector (including trade) and selected branches thereof. At the same time, the author/authors generalize the survey results and conclusions to the total Russian labor market, the overall Russian economy, the whole Russian society.

Such a temptation to overgeneralize the conclusions of the study of selected branches of the service sector only to the whole Russian economy and the total Russian labor market, or even the whole Russian society is due to the lack of necessary data presented earlier by the Reviewer from point 1) to point 8). The presentation of these data and their justification (e.g. very high share of the studied service sector branches in GDP, share in employment in selected service sector branches, in the whole Russian economy, etc.), may allow the Author/Author to generalize this research to the total Russian labor market, the whole Russian economy and the whole Russian society. On the basis of the reviewed text, the used data for selected branches of only the service sector of the Russian economy, the results of the study and the conclusions do not give grounds for such generalization of their conclusions to the total Russian labor market, the whole Russian economy and even to the whole Russian society.

Thus, for example, in the Conclusions (p. 16), the Author/Author presents that:

"The coronavirus pandemic contributed to an increase in the number of unemployed and had a decisive impact on the change in average wages."

(In fact, it should be noted that in selected branches of the service sector of the Russian economy) [1]

"The conclusions were obtained that in the conditions of wide informatization of society "

(In fact, it should be noted that informatization of society employed in selected branches of the service sector of the Russian economy)

" ... The study also established the rapid adaptation of the Russian society to the new conditions of remote employment." ...

(Actually, it must be pointed out, Russian society working in selected branches of the service sector of the Russian economy)

"... can contribute to the early stabilization of the Russian economy and the labor market from the consequences of COVID-19."

(Actually, it must be pointed out, Russian working society and the labor market in selected branches of the service sector of the Russian economy)

Yes, rapid adaptation of Russian society (p. 19) working in selected branches of the service sector of the Russian economy is possible to the new conditions of remote employment. However, is such rapid adaptation also possible in the broadly defined industrial sector (including: heavy industry, chemical, processing, food, construction and mining, etc.)? If not, the conclusions cannot be generalised to the whole Russian economy and the whole Russian labor market, or even to the whole Russian society.

[1] The reviewer uses italic font to indicate the necessary change in the conclusions of the study without presenting the data in items 1) -8). Presentation of these data and their justification (e.g. very high share of the studied service sector branches in GDP, share of employment of selected service sector branches in the whole Russian economy), may allow the Author/Author to generalize this research to the whole Russian labor market, the whole Russian economy and even to the whole Russian society.

II. The income of citizens cannot be equated with the average wages of employees (p.15).

"However, the study casts doubt on the statement that there is no direct effect of the pandemic on the average income of citizens: the analysis established a clear relationship between the spread of coronavirus infection and changes in the average wages of employees." (p. 15).

Citizens' income is a broader concept than employees' wages. First, income is a broader concept than wages. Second, citizen is a broader concept than employee.

After all, the article examines the regression of the average monthly wage of employees in selected branches of the service sector of the Russian economy. Again, the Author/Authors make a conclusion for the income of Russian citizens on this basis. The Author/Authors tend to over-generalize their research results and conclusions of selected branches of the service sector only to the whole Russian economy, the labor market of the whole Russian economy and the whole Russian society.

III. The dynamics of search queries on the Yandex search engine as presented in Table 1 (Table 1. Summary array of indicators, p. 4) from variable Y_(6-1) to variable Y_(6-6) are specific to selected branches of the service sector of the Russian economy. Which is even more strongly shown in Figure 1 (Figure 1. Conceptual research model, p. 5). In this Figure 1, the variable Y_(6-1) (remote work) influences behind the other 5 variables from Y_(2-6) to Y_(6-6). Remote work during the COVID-19 pandemic outside selected service sector branches is possible, however not in the whole Russian economy. At least this article does not prove it.

The adoption of such a conceptual research model contributes to the obtained only for selected service sector branches of the Russian economy. Figure 1 may correctly show the conceptual research model for selected service sector branches, but not for the whole Russian economy.

Overall, the article is important and interesting. However, it needs improvement at least in these three problematic areas. Maybe the title of the article needs to be changed: Impact of COVID-19 on the Russian service sector labour market: ... , instead of "IMPACT OF COVID-19 ON THE RUSSIAN LABOR MARKET: ... Of course, in this case it is worth justifying that the service sector is important in the Russian economy and also the selected service sector industries studied, are important enough in the Russian economy, based on the data in the table or tables from point 1) to 8) indicated by the Reviewer. I hope that my review will be perceived as constructive and serving to improve this article.

Author Response

We are truly grateful for your profound and constructive review on the article!

In fact, we should admit that the lack of necessary information on the scale and scope of the selected sector and branches in the Russian economy may mislead the reader, which would consequently lead to giving him a wrong idea of overall significance of the study. Thus, following your advice and in order to avoid misinterpretations, we have added official statistics from Russian Federal State Statistics Service on the share of the selected sector in the GDP of Russia as well as the share of average annual number of employed in the selected branches of the Russian economy (Figures 1-2). According to Russian Federal State Statistics, the selected sector makes up more than half of the GDP of the Russian economy, in addition, more than a third of the able-bodied population of the Russian economy is employed in selected service sector branches, therefore, we believe that it gives the grounds for confirming the significance of the study to the whole Russian labor market and, consequently, the title of the article in its current version.

The rationale of the economy branches, taken into consideration within the framework of the study, has also been added to the paper. The information is now spelled out in "Introduction" section.

By the way, we should admit that the income of citizens indeed cannot be equated with the average wages of employees – therefore, we have made appropriate edits, highlighting the results, obtained during the study.

Speaking of the dynamics of search queries on the Yandex search engine, we should make additional explanations. In Figure 3 (Figure 1 in previous version of the article), we present conceptual research model, which implies multiple regression and paired regression equations. Within the framework of the study we pay particular attention to the block devoted to the study of the dynamics of changes in related search queries regarding the search query "remote work" (paired regression). The study of these dependencies indirectly allowed to reveal the logic of the behavior of the Russian society during the spread of the new coronavirus infection. Of course, it cannot be said that the study of paired regressions fully describes the logic of people behavior during the pandemic (to establish such connections, it is necessary to analyze a much larger array of search queries), nevertheless, the results of the paired regression analysis, related to the selected service sector branches of the Russian economy, allowed to draw conclusions on how much society was ready for the changes that took place, to what extent citizens were ready to change their behavior patterns in a crisis. Meanwhile multiple regression, helped us to draw conclusions on the specificity of the impact of COVID-19 on selected branches of the economy.

We take this opportunity of thanking you one more time for your attentive assistance and contributing so much to enhancing the study!

Reviewer 5 Report

-the abstract is too long. It is suggested to make it leaner

-The problem, i.e. the gap in the literature, needs to be better presented in the introduction.  The research question and objectives are not well understood.

-If possible, create a paragraph or sub-paragraph of the introduction in which the literature review is stated. This will make the article more readable.

-I suggest inserting a sentence explaining how the article is structured.

-There are no references to the limits of the research/methodology and possible future research.

Author Response

We are truly grateful for your feedback on the article and the advice given to strengthen it!

Following your advice, we have made the abstract leaner, now it really looks more readable. Moreover, we have revised the structure of "Introduction" section: the gap in the literature is now better presented as well as the research question and objectives, moreover there is a part of the text there, in which the literature review is stated. In the end of the "Introduction" section there is now a paragraph, explaining the way the study is structured.

The references to the limits of the research and possible future research have been also added in the "Conclusions" section.

We take this opportunity of thanking you one more time for your attentive assistance and contributing so much to enhancing the study!

Reviewer 6 Report

Referee Report for the article “Impact of COVID-19 on the Russian labor market: comparative analysis of the physical and informational spread of the coronavirus”
This article jointly models the association between the physical and informational spread of COVID-19 and macroeconomic outcomes in the Russian labour market. The focus on the additional dimension of the information flow is interesting and this article could potentially be an interesting contribution to the international literature. However, in its current form there several major flaws which need to be addressed to obtain reliable results.
(1) My main concerns relate to the methodology which is not explained well and appears to have a number of flaws:
(a) Based on line 179, it seems that the paper deals with monthly time-series of macroeconomic data and COVID-19 related statistics. However, there is no discussion of basic time series concepts and data transformations. Have the authors checked for stationarity (trends, seasonality, unit roots), autocorrelation structure of the dependent variable and the error term, etc.? Are the estimated standard errors robust? Are macroeconomic variables transformed (changes / log changes) or are the authors dealing with raw data? If not, the regressions are not correctly specified and may be spurious and provide misleading results.
(b) How were the variables selected that appear in Table 1? Is unemployment available on industry level? It is likely that COVID-19 did not have the same impact on the unemployment across different industries. The measures (units?) and the sources in Table 1 need to be spelled out as it is difficult to follow in its current form.
(c) How was the sample period (January-August 2020) selected? Some background about COVID-19 in Russia, policies and restrictions is needed for readers outside of Russia to understand the context. The authors mention three stages in lines 177-179 but there are no corresponding dates, and this does not appear to be explicitly modelled later on. The authors mention that the analysis is done in monthly frequency (see line 179) which would mean that there are only 8 observations per variable? Why do the authors not include more data prior to the pandemic and during the pandemic?
(d) Table 2 is confusing. What is ?22? Is it ?2 squared? There is no mention of that in the methodology section. It would also be helpful to have meaningful abbreviations of variables in the presentation of results (for example, separate columns for each variable).
(e) The regressions appear to have been estimated via OLS, but there is no mention of the significance of the estimated coefficients. There is no clear labelling regarding the significance of each estimated coefficient! There is a column called “P-level”. Is it the p-value for the F-test? The statistical significance for each estimated coefficient must be given in some form (for example, t-statistics, standard errors, confidence interval, p-values or *, **, *** indicating significance at 10%, 5% or 1%, respectively). We cannot interpret or draw any conclusions without knowing the statistical significance of each individual estimated coefficient.
(f) The methodology does not deal with potential endogeneity, such as reverse causality. Is OLS appropriate in this context? Have the authors considered any approaches to deal with endogeneity or any alternative methodologies such as VAR models?
(g) There could be a number of confounding factors, including government policies, lockdowns and restrictions. Is it really the number of COVID-19 cases that drives unemployment? There is probably a much more complex mechanism, which is likely to be related to government policies, lockdowns, border closures, school closures, and perhaps some other factors as well. This needs to be discussed and modelled explicitly.
(h) There are no descriptive statistics or graphical illustrations of the time series variables for their sample period. It would also be helpful to see a comparison before and during the pandemic.
(i) Further, throughout the manuscript there are references to a causal relationship (e.g., “to test the hypothesis that physical and informational spread of coronavirus caused an increase in the number of unemployed, (…)” in lines 13-14), however in its current form the analysis looks at associations between variables rather than causal links.
(2) What is the significance and main contribution of this work? How is it relevant outside of Russia? How does it fit within the large body of literature on the impacts of COVID-19 on international labour markets? I would suggest emphasising the importance of modelling information flows, build on that and link that to the broader literature.
(3) There are multiple hypotheses in this paper that are subsequently tested. While there is some related literature in the introduction, I would suggest connecting the literature more explicitly to the hypotheses. For example, each hypothesis can be stated in a separate paragraph and then backed up with relevant theory and empirical findings in the international literature.
(4) The article needs to be professionally edited by a native English speaker.

Author Response

We are truly grateful for your feedback on the article and the advice given to strengthen it!

To begin with, we should emphasize the research methodology, which is based on the traditional regression analysis. We implied such defining criteria as: the presence of structural breaks / structural outliers, significance of Fisher's F-test, P-level, approximation error of the model, multiple coefficient of determination R2, heteroscedasticity of residues, the presence of multicollinearity, analysis of the relative coefficients of elasticity. Moreover, we have checked for stationarity, autocorrelation structure of the dependent variable and the error term. The estimated standard errors turned out to be robust. Obviously, the study deals with transformed data, the corresponding explanations are given in "Materials and methods" section. In the study we didn’t go into too much formal details, since it would significantly overload the paper and consequently decrease its readability.

Besides, current version of the article includes leaner presentation of the regression results with their meaningful interpretation just under Table 4.

We should admit that the lack of the sources and the measures explanation indeed made the paper a bit confusing. Following your recommendations, we spell it out now in the current version of the article. The list of the sources used within the framework of the study is now described in the very beginning of "Materials and methods" section. The abbreviations used for measures in Tables 1-3 are described just above Table 1. We empathize that the variables selection is driven by the analysis of the theoretical and methodological basis, which is now thoroughly described in "Introduction" section. We also outline the HeadHunter Research Center’s (the largest online recruitment platform in Russia) investigation on the most and least affected sectors of Russian economics because of the coronavirus pandemic. The study showed that the service sector (including trading) was affected most because of self-isolation restrictions and physical incapability of selling goods and services to the customers, which determined the logic of our study.  Anyway, we believe that the additional edits contributed to the study’s improvement.

Talking about the framework of the study, the temporary array of statistical data is really divided into three stages, corresponding the stages of coronavirus spread and the governmental measures to prevent it (the primary data is now available in Table 1). The first stage is the growth of infection distribution and the absence of any governmental measures to prevent the further spread; the second stage is the peak of distribution, accompanied by severe governmental restrictions – a period of "self-isolation" for citizens, which obliged people not to leave their homes without any crucial reasons, compulsory closure of trade and service enterprises except for those selling living essentials; the third stage is the decline in the spread of COVID-19 and the mitigation of substantial part of the governmental restrictions for citizens and business. The paper examines the adaptation of Russian society to the rapidly changed conditions of the working environment during the most distinct restrictive governmental measures related to the COVID-19. We stress that the forced introduction of interaction restrictions led to a significant transformation of the production and consumption processes, which resulted in the disruption of many economic ties, whereas further “waves” of the COVID-19 were accompanied by more formal governmental restrictive measures, followed by Russian society to a comparatively lesser extent. The investigation of this timeframe (after August 2020) does not fully meet the original goals of the study. Anyway, the variables selected for the purposes of the study are complex and therefore reflect the influence of many factors. We fully agree that the lack of additional information about COVID-19 background in Russia hampered the study’s understanding for readers outside of Russia. Thus, we have added descriptive statistics and primary data (Tables 1-2) in "Materials and methods" section.

?22 is indeed ?2 squared. The methodology of the study includes the assumed square, cube connection in the description of the system of equations.

You are right, stressing the possibility of a number of confounding factors to drive unemployment. The reason we didn’t include more complex mechanism is the original purpose of the study, which examines we the impact of COVID-19 on the list of economic indicators. Anyway, we have added the corresponding remark to the limits of the study. Unfortunately, unemployment data is still unavailable at the industry level, in the future we will update the data when the statistics will appear.

Moving on, in order to better present the research question and objectives in the literature review, the structure of "Introduction" section has been transformed, now the literature is connected to the hypotheses more explicitly.

By the way, your review led us to the idea that the significance of the study should be more clearly outlined. The study confirms the coronavirus physical spread direct impact on the elements of the Russian labor market, informational spread indirect impact on the transformation of the external environment, as well as the behavior of society, which leads to the necessity of modeling information flows to minimize negative economic effects on the Russian labor market. That means that a similar influence may take place on international labor markets, which determines possibility to apply the results of this study by authors outside Russia. The information is now spelled out in “Conclusions” section.

In addition, we take your advice on professional edition of the paper into consideration, since it would really strengthen the study.

We take this opportunity of thanking you one more time for your attentive assistance and contributing so much to improving the article!

Round 2

Reviewer 2 Report

The re-submitted paper addresses the impact of COVID-19 pandemic on the Russian labor market. The authors have substantially improved the paper. However, there are some grave issues that have to be addressed. The central shortcoming is the fact that the author has not addressed the specifics of the labor market of a resource-rich and resource exporting transition economies. These are economies dominated by oil, gas and ore exports, which are not strongly linked to the domestic economy. To fix this issue and to provide the state of the art, refer to the following studies on Azerbaijan and Kazakhstan. These are https://doi.org/10.1007/s13563-019-00202-6   and https://doi.org/10.1080/14631377.2020.1745557

Author Response

Dear reviewer, we are extremely grateful both for your assessment of our work and for the recommendation to refer to these articles. The research results presented in them are truly impressive and useful for our research. We referenced them in our article. Once again, thank you very much!

Reviewer 3 Report

I appreciate all the major revisions that the authors have made.

Author Response

Dear reviewer, we are extremely grateful for your assessment of our work.

Reviewer 4 Report

I appreciate the changes made by the Authors. After the improvement of the article is its content with the calculations made is quite coherent, understandable and clear. However, the improvement of several errors is still required.

The overall merit of the reviewed article I assess highly, assuming that during the revision of the article several very important errors were not noticed. Therefore , because of these serious errors, which affect: presentation of results, quality of presentation, support of conclusions by the results of the study, they should be corrected. After correcting them I assess the article highly. I believe that the authors will correct these apparent simple errors that affect my low rating of only three points in the review form. Once the errors have been corrected, I rate the article highly.

Several errors below need to be corrected in the article.

Under Table 1 or under Table 3 in the explanatory notes, the concept or formula Workload of unemployed population per 100 announced vacancies needs to be clarified. This is important for the Reader because from April to August its values are already above 100 and they are constantly growing, "pax" is for persons, which is evident in the analysis made and even more strongly confirms the formulated conclusions.

On p. 4 (line 217), there is an own name of the Internet platform HeadHunter Research Center or on p. 5 (line 250) there is also an own name RBC (RosBusinessConsulting). In the article I suggest to write proper names in italic letters or to distinguish them in another uniform way.

On p. 4 (Figure 4. Validated conceptual model) between the variable y_(6-1) and the variable y_(6-3) there is a value as high as 70. Please be sure to check for a serious error here. If the value of 70 is correct, please substantively justify this exceptional and very high value, compared to other values in Figure 4 and other calculations on these variables in the article.

On page 5 (line 234) it is necessary to change the title of Figure 2. It refers to the branch of the service sector only, and not to the overall Russian economy. This mistake is very important and it has very serious implications for the analysis made and conclusions formulated. The content of the article from line 228-229 (p. 4) cannot be different from line 234 on p. 5. I propose the title Figure 2. Share of average annual number of employed in the service sector branches of the Russian economy (p. 5, line 234), instead of Figure 2. Share of average annual number of employed in the branches of the Russian economy (p. 5, line 234).

It is necessary to improve the numbering in the first column of Table 3 (from line 272 downwards). The shifted numbering in Table 3 creates disorder and lack of clarity.

On p. 7 (line 769) there is an error in the references. Is "[2] Tisdell, C.A.. (2020). ... " is correctly spelled in this line?

I look forward to the published revised article.

Author Response

Dear reviewer, we are extremely grateful for your assessment of our work, as well as for your comments. By checking them, our article has become much more representative and scientifically significant. In accordance with your review, we have done the following corrections:

  1. We supplemented the text of the article with explanations regarding the workload of the unemployed population per 100 announced vacancies.
  2. We have adjusted the presentation of the names of the analyzed Internet platforms.
  3. In Figure 4, there was indeed a gross error. Thank you very much, we have fixed it.
  4. We have included in the text a set of explanations regarding the significance of the study and the results obtained specifically for the service sector.
  5. We have adjusted the numbering in Table 3.
  6. We have adjusted the list of references.

Once again, thank you very much!

Reviewer 5 Report

Now it's ok!

Author Response

(The authors gave the same response as above.)

Reviewer 6 Report

The authors have clarified many of the questions and issues and provided more background information. However, there are still areas that have not been addressed sufficiently:

(1) The presentation of results in Table 4 is still inappropriate. There is still no indication of the statistical significance of each coefficient (in the first column of Table 4). My comment from the previous review round has not been addressed: 

"The statistical significance for each estimated coefficient must be given in some form (for example, t-statistics, standard errors, confidence interval, p-values or *, **, *** indicating significance at 10%, 5% or 1%, respectively). We cannot interpret or draw any conclusions without knowing the statistical significance of each individual estimated coefficient." 

I have included a link to a recent Economies publication (Fendel et al., 2022) to illustrate what I mean: 

https://www.mdpi.com/2227-7099/10/5/109/htm 

For example, their Table 3 presents different specifications in columns (1) through (4), lists each regressor in the first column (spelling out the variable instead of using abbreviations such as X), and indicates statistical significance for each coefficient (*, **, or ***).

I suggest changing the format of your Table 4 to a similar format as their Table 3. To save space, you can remove some columns, for example "heteroscedasticity of residuals" and add a note below the table. Then you could say in the note that potential heteroscedasticity is observed in equations that are labelled as ^ (or another suitable symbol). 

(2) Could the authors please clarify their response further?

“The estimated standard errors turned out to be robust. Obviously, the study deals with transformed data, the corresponding explanations are given in "Materials and methods" section. In the study we didn’t go into too much formal details, since it would significantly overload the paper and consequently decrease its readability.”

First, could you mention which standard errors you use (e.g., errors that are robust to heteroscedasticity / autocorrelation, clustered errors, etc.)? For example, Table 3 from the link that I shared above (Fendel et al., 2022) has a note saying that “Standard errors clustered at industry level are reported in parenthesis” to clarify this issue. Since you mention in Table 4 that you identify heteroscedasticity in some of your specifications, it is important to use heteroscedasticity robust standard errors in this case. If you also have autocorrelation, you can use HAC standard errors. In any case, the type of standard error that you choose needs to be mentioned explicitly.

Second, could you elaborate on “transformed data”? What transformations have you used? For example, is the unemployment data (Y1) expressed in levels, changes (i.e., first differences), percentage changes, log changes, …? I couldn’t find this in the “Materials and methods” section. My concern is about the possibility of spurious regressions when time-series variables contain a stochastic trend but this is not addressed by transforming the variable… Most importantly, you need to reassure the reader that you have checked for stochastic trends (e.g., via the Dickey-Fuller tests) and in case there were any stochastic trends you need to show that you have transformed the variable accordingly (e.g., taken first differences). Then, you could add this information to Table 3 in your column named “Measure” and mention whether each variable is in levels, first differences, or other transformations.

(3) Thanks for confirming the following:

“?22 is indeed ?2 squared. The methodology of the study includes the assumed square, cube connection in the description of the system of equations.”

This explanation needs to be included in your paper when you introduce the equations. For example, this can be similar to the format in Equation (1) in Fender et al. (2022) (this is the paper from the link above).

 (4) I still suggest professional editing of the paper. Just as an example, there is a grammatical error in the sentence in line 263 “The summary statistics is presented in Tables 1-2” instead of “The summary statistics are presented in Tables 1-2”.

Author Response

Dear reviewer, we are extremely grateful for your assessment of our work and such detailed comments. We will answer them in sequence:

  1. Indeed, we did not take into account the need to indicate the significance of each variable in our equations. We have added this information to Table 4. We've also added a comment about heteroscedasticity in the form you suggested.
  2. As tools for estimating standard errors, we used Heteroskedasticity and Autocorrelation consistent standard errors, which we also noted in the article. Regarding “transformed data”, we apologize, in the framework of the previous explanation, we meant that there was no need for it We checked for the presence of stochastic trends in the time series, and did not define it.
  3. Thank you for your comment regarding the quadratic and cubic relationship information. We have added the related comments to the text of the article.
  4. We are extremely grateful for your recommendation regarding the professional proofreading of the article. We used the services of specialists from our university, and they corrected possible lexical, spelling and punctuation inaccuracies.

Once again, thank you very much!

Round 3

Reviewer 6 Report

Thanks for your response and I wish you all the best with your paper.

This manuscript is a resubmission of an earlier submission. The following is a list of the peer review reports and author responses from that submission.

Round 1

Reviewer 1 Report

Summary

This paper addresses the impact of the COVID-19 pandemic on the Russian labor market. The authors differentiate between informational and physical impact. The informational impact is measured by the frequency of news, in the RBC Vkontakte news group, related to the topic “coronavirus”. In turn, the physical impact is measured by the monthly spread of coronavirus. The authors follow a multiple regression analysis approach. Most of the hypothesized relationships were confirmed by the analysis.

General comment

The topic of this paper is timely. Nevertheless, the approach followed by the authors does not present any theoretical or methodological contribution for publication in Mathematics. I recommend the authors improve the paper and submit it to a more suitable outlet.

Specific comments

Comment #1

It is unclear why the authors have considered the period between January to August 2020. Why ending in August 2020 and consequently only consider eight months to perform the analysis?

Comment #2

The regression equations presented in page 6 are not numbered. Moreover, the functional forms presented do not match the ones presented in Table 2. For example, on page 6 we have $y_1= a + b_1X_1 + b_2X_2$ and in Table 2 we observe that the regression estimated corresponds to $y_1 = a + b_1X_1 + b_2X_2 + b_3X_1^2 + b_4X_1^3$. The estimated functional forms should be explained in more detail.

Comment #3

The choice of a 10% significant level is at least debatable. Furthermore, on page 7, we can read, “Thus, every time the P-level of any Y exceeds the acceptable level of 10%, the multiple regression model is being optimized”. Does this mean that the authors have tested multiple models until achieving the one with a p-value less than 10%?

Comment #4

What were the performed tests to identify structural breaks/outliers? How was the heteroscedasticity of the residuals assessed?

Reviewer 2 Report

Dear authors,

The paper does not correspond (it does not match) in content to the aim of the special issue. The approach is not addressing in the way of develop and implement advanced mathematical and instrumental methods in the field of assessing resilience in various economic and business sectors.

Overall, the paper have a lot of weaknesses:

  1. First of all is insufficiently represented by a comprehensive literature review (the references list contain only 13 titles!!!);
  2. The chosen period is not at all representative (January-August 2020). The COVID-19 pandemic began to generate infections registered by the authorities, at the earliest in March 2020. Why is it being studied in January and February? Why is August 2020 taken into account as the final month? Significant increases in cases of COVID-19 virus infection have occurred in the last three to four months of 2020, and major labor market influences have manifested themselves worldwide since October 2020.
  3. Table 1 in the paper mentions some indicators but is mentioned in most cases the numbered source [11]. Why would this work be representative? Why would it be so useful? A simple example: among the types of economic activities taken into account there is not a very strong sector affected by the pandemic (such as the transport sector; the air transport sector had the biggest losses in terms of the volume of people transported !!!). 
  4. For the relevance and usefulness of the research it is necessary to take into account other studies that have proposed some indicators (for example in the case of the resilience programs of the EU member states various indicators have been considered).
  5. The research methodology chosen by the authors is already classical (based on regression modeling), and the question is why the authors chose this tool as the method of analysis? What were the reasons for choosing the regression? There are some materials that confirm or sustain as being appropriate to use regression modeling?  
  6. What would be the benefits / advantages of a comparative analysis of the physical and informational spread of coronavirus because the responsibility for the measures taken depends on the authorities (central and local), and these solutions cannot be replicated in different places or countries in the same way (eg China, United States, Germany, France, Australia, Canada, etc.) and cannot be exploited for the purpose of treating them as good practices to be applied. Specifically, what can this study really help? I do not see useful things for interested readers or organizations! I do not see an essential contribution of the findings described in this paper to this field!
  7. The paper has many limitations (which are not described) and many shortcomings (through the considered interval, through the considered economic fields, by substantiating an adequate research methodology etc.). Therefore, I encourage the authors to consider all these aspects mentioned above and to resubmit with a fresh paper, containing an improved work, that also corresponds to the aims of special issues of the journal! 
  8. Basically, what is now included in the paper is a good start to redesign and rewrite the article!

Reviewer 3 Report

The topic of the impact of the Covid pandemic on the national economy is currently strong and strong, the content of the article fits into a special issue of the Mathematics journal.

The authors set two goals for the work, but basically dealt with only one - to investigate the impact of the new coronavirus on the Russiona labor market. The second goal - suppose the actions to be taken to minimize negative economic consequences - remained almost unresolved and is mentioned only in general and is not based on data and results of the work.

I have reservations about the structure of the work - it is not customary that in scientific work the Objectives of the work are mentioned in the Introduction in various places (for example line / row 63 ..., 70 ..., 148 ... etc.) I suggest to be followed by a special subchapter 2. Objectives of the study behind the theoretical part in Introduction.

(Formally – article requires minor language corrections – for example line 158-165 (8 rows) is just one sentence, it is very difficult to read it and to be well oriented within the text and authors argue).

The Material and Methods section is acceptable, the authors use data from January to August 2020 and use the informal Russian social network Vkontakte. Why - an unofficial - source Vkontakte and do not use official state or regional data? Can they explain and defend this?

I have no comments on own statistical and mathematical processing. The validated conceptual model is described by multiple and paired regression equations. The results are clearly processed into tables and are verbally commented. This only applies to the first three pages of Resluts after line about 445.

From line 446, the authors discuss the Results and some conclusions - in the form of a discussion - are not proven. Discussion is a standard part of scientific articles, however, it is missing here. I propose to change the structure of Results, or even better, to insert (after Results) from line about 446… subchapter Discussion, which is more suitable for the text.

The work is generally of standard quality, but the Half-Page Conclusions are severely inadequate and insufficient. Also the second goal - suppose the actions to be taken to minimize negative economic consequences is very short – it would be probably better to exclude this part out of the work. Or – if authors will insist on it, make it stronger and based on Dta and work Results.